# Offline Model-Based Optimization by Learning to Rank

**Rong-Xi Tan**[1,2]**, Ke Xue**[1,2] **, Shen-Huan Lyu**[3,4,1] **, Haopu Shang**[1,2]
**Yao Wang**[5] **, Yaoyuan Wang**[5] **, Sheng Fu**[5] **, Chao Qian**[1,2*]
[1] National Key Laboratory for Novel Software Technology, Nanjing University, China
[2] School of Artificial Intelligence, Nanjing University, China
[3] Key Laboratory of Water Big Data Technology of Ministry of Water Resources, Hohai University, China
[4] College of Computer Science and Software Engineering, Hohai University, China
[5] Advanced Computing and Storage Lab, Huawei Technologies Co., Ltd., China

## Abstract

Offline model-based optimization (MBO) aims to identify a design that maximizes a black-box function using only a fixed, pre-collected dataset of designs and their corresponding scores. This problem has garnered significant attention from both scientific and industrial domains. A common approach in offline MBO is to train a regression-based surrogate model by minimizing mean squared error (MSE) and then find the best design within this surrogate model by different optimizers (e.g., gradient ascent). However, a critical challenge is the risk of out-of-distribution errors, i.e., the surrogate model may typically overestimate the scores and mislead the optimizers into suboptimal regions. Prior works have attempted to address this issue in various ways, such as using regularization techniques and ensemble learning to enhance the robustness of the model, but it still remains. In this paper, we argue that regression models trained with MSE are not well-aligned with the primary goal of offline MBO, which is to *select* promising designs rather than to predict their scores precisely. Notably, if a surrogate model can maintain the order of candidate designs based on their relative score relationships, it can produce the best designs even without precise predictions. To validate it, we conduct experiments to compare the relationship between the quality of the final designs and MSE, finding that the correlation is really very weak. In contrast, a metric that measures order-maintaining quality shows a significantly stronger correlation. Based on this observation, we propose learning a ranking-based model that leverages learning to rank techniques to prioritize promising designs based on their relative scores. We show that the generalization error on ranking loss can be well bounded. Empirical results across diverse tasks demonstrate the superior performance of our proposed ranking-based method than twenty existing methods. Our implementation is available at `https://github.com/lamda-bbo/Offline-RaM`.

## 1 Introduction

The task of creating new designs to optimize specific properties represents a significant challenge across scientific and industrial domains, including real-world engineering design (Kumar et al., 2022; Shi et al., 2023), protein design (Khan et al., 2023; Kolli, 2023; Chen et al., 2023b; Kim et al., 2023), and molecule design (Gaulton et al., 2012; Stanton et al., 2022). Numerous methods facilitate the generation of new designs by iteratively querying an unknown objective function that correlates a design with its property score. Nonetheless, in practical scenarios, the evaluation of the objective function can be time-consuming, costly, or even pose safety risks (Dara et al., 2022). To identify the next candidate design using only accumulated data, offline model-based optimization (MBO; Trabucco et al., 2022) has emerged as a widely adopted approach. This method restricts access to an offline dataset and does not allow for iterative online evaluation, which, however,

---
*Correspondence to Chao Qian <qianc@nju.edu.cn>

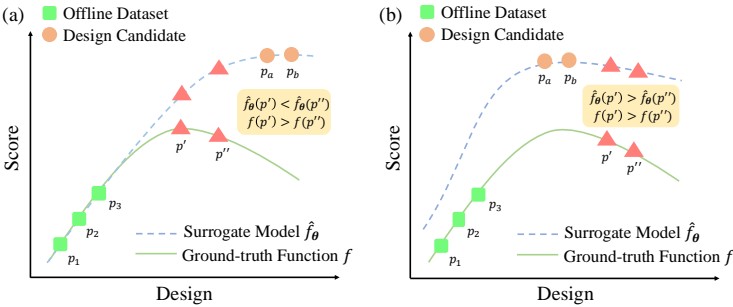

Figure 1: Illustration of (a) OOD issue of regression-based models and (b) order-preserving ranking-based models. In (a), the regression-based method searches into suboptimal regions. Prior works focus on high OOD-MSE, while in this work, we point out that it is caused by the OOD error in preserving order. In (b), although the surrogate model also has high OOD-MSE, it can maintain the order, thus resulting in good design candidates.

also results in significant challenges. A common strategy, referred to as the *forward* method, entails the development of a regression-based surrogate model by minimizing mean squared error (MSE), which is subsequently utilized to identify the optimal designs by various ways (e.g., gradient ascent).

The main challenge of offline MBO is the risk of out-of-distribution (OOD) errors (Kim et al., 2025), i.e., the scores in OOD regions may be overestimated and mislead the gradient-ascent optimizer into suboptimal regions, as shown in Figure 1(a). Thus, overcoming the OOD issue has been the focus of recent works, such as using regularization techniques (Trabucco et al., 2021; Fu & Levine, 2021; Yu et al., 2021; Chen et al., 2022; Qi et al., 2022; Dao et al., 2024b) and ensemble learning (Yuan et al., 2023; Chen et al., 2023a) to enhance the robustness of the model, but it still remains.

Recent studies (Hoang et al., 2024) have pointed out that value matching alone is inadequate for offline MBO. In this paper, we conduct a more thorough and systematic analysis on this view. We aim to answer the key question: "Is MSE a good metric for offline MBO?" Consequently, we find through experiments that the relationship between the quality of the final designs and MSE in the OOD region (denoted as OOD-MSE) is weak, which underscores the need for a more reliable evaluation metric.

Next, we reconsider the primary goal of offline MBO, which seeks to identify the optimal design $\mathbf{x}^*$ over the entire design space. Intuitively, this process does not require exact score predictions from the surrogate model; rather, it demands that the model accurately discerns the partial ordering of designs. As shown in Figure 1(b), if a surrogate model can maintain the order of candidate designs based on their relative score relationships, it can produce the best designs even without precise predictions. We prove the equivalence of optima for order-preserving surrogates, and introduce a ranking-related metric, Area Under the Precision-Coverage Curve (AUPCC), for offline MBO, which shows a significantly stronger correlation with the final performance.

Based on this observation, we propose learning a **Ra**nking-based **M**odel (RaM) that leverages learning to rank (LTR) techniques to prioritize promising designs based on their relative scores. Our proposed method has three components: 1) *data augmentation* to make the offline dataset align with LTR techniques; 2) *LTR loss learning* to train the RaM; 3) *output adaptation* to make gradient ascent optimizers work well in RaM. We show that the generalization error on ranking loss can be well bounded, and conduct experiments on the widely used benchmark Design-Bench (Trabucco et al., 2022). Equipped with two popular ranking losses, i.e., RankCosine (Qin et al., 2008) and ListNet (Cao et al., 2007), our proposed method, RaM, performs better than state-of-the-art offline MBO methods. Ablation studies highlight the effectiveness of the main modules of RaM. We also examine the influence of different ranking loss, and demonstrate the versatility of ranking loss, bringing improvement even by simply replacing the MSE loss of existing methods with ranking loss.

The contributions of this work are highlighted in three key points:

1) To the best of our knowledge, we are the first to indicate that MSE is not suitable for offline MBO.

2) We show that the ranking-related metric AUPCC is well-aligned with the primary goal of offline MBO, and propose a ranking-based method for offline MBO.

3) We conduct comprehensive experiments across diverse tasks, showing the superiority of our proposed ranking-based method over a large variety of state-of-the-art offline MBO methods.

## 2 BACKGROUND

### 2.1 OFFLINE MODEL-BASED OPTIMIZATION

Given the design space $\mathcal{X} \subseteq \mathbb{R}^d$, where $d$ is the design dimension, offline MBO (Trabucco et al., 2022; Kim et al., 2025; Qian et al., 2025; Xue et al., 2024) aims to find a design $\mathbf{x}^*$ that maximizes a black-box objective function $f$, i.e., $\mathbf{x}^* = \arg\max_{\mathbf{x} \in \mathcal{X}} f(\mathbf{x})$, using only a pre-collected offline dataset $\mathcal{D}$, without access to online evaluations. That is, an offline MBO algorithm is provided only access to the static dataset $\mathcal{D} = \{(\mathbf{x}_i, y_i)\}_{i=1}^N$, where $\mathbf{x}_i$ represents a specific design (e.g., a superconductor material), and $y_i = f(\mathbf{x}_i)$ represents the target property score that needs to be maximized (e.g., the critical temperature of the superconductor material).

The mainstream approach for offline MBO is the *forward* approach, which fits a surrogate model, typically a deep neural network $\hat{f}_{\boldsymbol{\theta}} : \mathcal{X} \rightarrow \mathbb{R}$, parameterized by $\boldsymbol{\theta}$, to approximate the objective function $f$ in a supervised manner. Prior works (Trabucco et al., 2021; Fu & Levine, 2021; Yu et al., 2021; Qi et al., 2022; Yuan et al., 2023; Chen et al., 2023a; Hoang et al., 2024; Dao et al., 2024b) learn the surrogate model by minimizing MSE between the predictions and the true scores:

$$\arg\min_{\boldsymbol{\theta}} \sum_{i=1}^N \left( \hat{f}_{\boldsymbol{\theta}}(\mathbf{x}_i) - y_i \right)^2 / N.$$

With the trained model $\hat{f}_{\boldsymbol{\theta}}$, the final design can be obtained by various ways, typically gradient ascent:

$$\mathbf{x}_{t+1} = \mathbf{x}_t + \eta \left. \nabla_{\mathbf{x}} \hat{f}_{\boldsymbol{\theta}}(\mathbf{x}) \right|_{\mathbf{x}=\mathbf{x}_t}, \quad \text{for } t \in \{0, 1, \dots, T-1\}, \tag{1}$$

where $\eta$ is the search step size, $T$ is the number of steps, and $\mathbf{x}_T$ serves as the final design candidate to output. However, this method is limited by its poor performance in out-of-distribution (OOD) regions, where the surrogate model $\hat{f}_{\boldsymbol{\theta}}$ may erroneously overestimate objective scores and mislead the gradient-ascent optimizer into sub-optimal regions. There have been many recent efforts devoted to addressing this issue, such as using regularization techniques (Fu & Levine, 2021; Trabucco et al., 2021; Yu et al., 2021; Dao et al., 2024b;a) and ensemble learning (Yuan et al., 2023; Chen et al., 2023a) to enhance the robustness of the model.

Another type of approach for offline MBO is the *backward* approach, which typically involves training a conditioned generative model $p_{\boldsymbol{\theta}}(\mathbf{x}|y)$ and sampling from it conditioned on a high score. For example, MINs (Kumar & Levine, 2020) trains an inverse mapping using a conditioned GAN-like model (Goodfellow et al., 2014); DDOM (Krishnamoorthy et al., 2023) directly parameterizes the inverse mapping with a conditioned diffusion model (Ho et al., 2020); BONET (Mashkaria et al., 2023) uses trajectories to train an autoregressive model, and samples them using a heuristic.

A comprehensive review of offline MBO methods is provided in Appendix A.1 due to space limitation. In this paper, we point out that the regression-based models trained with MSE are not well-aligned with offline MBO's primary goal, which is to *select promising designs* rather than predict exact scores. Intuitively, offline MBO does not require exact score predictions from the surrogate model; rather, it demands that the model accurately discerns the partial ordering of designs, which naturally aligns with the learning to rank (LTR) framework introduced in Section 2.2.

### 2.2 LEARNING TO RANK

LTR aims to learn an optimal ordering for a given set of objects (e.g., designs in offline MBO), and has applications across various domains, including information retrieval (Liu, 2010; Li, 2011), recommendation systems (Karatzoglou et al., 2013), and language model alignment (Song et al., 2024; Liu et al., 2024). It is typically formulated as a supervised learning task. Given the training data $\mathcal{D}_{\mathrm{R}} = \{(\mathbf{X}, \mathbf{y}) \mid (\mathbf{X}, \mathbf{y}) \in \mathcal{X}^m \times \mathbb{R}^m\}$, where $\mathcal{X}$ is the object space, $\mathbf{X}$ is a list of $n$ objects to be ranked, each denoted by $\mathbf{x}_i \in \mathcal{X}$, and $\mathbf{y}$ is a list of $n$ corresponding relevance labels $y_i \in \mathbb{R}$, the goal of LTR is to learn a ranking function that assigns scores to individual objects and then arranges these scores in descending order to produce a ranking. Formally, LTR aims to identify a ranking score function $s_{\boldsymbol{\theta}} : \mathcal{X} \rightarrow \mathbb{R}$, parameterized by $\boldsymbol{\theta}$. Let $s_{\boldsymbol{\theta}}(\mathbf{X}) = [s_{\boldsymbol{\theta}}(\mathbf{x}_1), s_{\boldsymbol{\theta}}(\mathbf{x}_2), \dots, s_{\boldsymbol{\theta}}(\mathbf{x}_m)]^\top$, and we can optimize the model by minimizing the empirical loss:

$$\mathcal{L}(s_{\boldsymbol{\theta}}) = \sum_{(\mathbf{X}, \mathbf{y}) \in \mathcal{D}_{\mathrm{R}}} l\left(\mathbf{y}, s_{\boldsymbol{\theta}}(\mathbf{X})\right) / |\mathcal{D}_{\mathrm{R}}|,$$

where $l(\cdot)$ is the loss function applied to each list of labels and predictions. Depending on their approach to handling ranking loss, LTR algorithms are categorized into three types: 1) *Pointwise* (Crammer & Singer, 2001): Treat ranking as a regression or classification problem on individual objects; 2) *Pairwise* (Köppel et al., 2019): Transform ranking into a binary classification problem on object pairs; 3) *Listwise* (Xia et al., 2008): Directly optimize the ranking of the entire list of objects.

## 3 METHOD

In this section, we introduce our ranking-based surrogate models for offline MBO. We first analyze in detail the goal of offline MBO and aim to answer the critical question, "Is MSE a good metric for offline MBO?" in Section 3.1. Consequently, we find that MSE is not a suitable metric, and thus introduce a better one in Section 3.2, i.e., Area Under the Precision-Coverage Curve (AUPCC), which is related to ranking. This motivates us to propose a framework based on LTR to solve offline MBO in Section 3.3. Furthermore, we show that the surrogate model based on LTR methods can have a good generalization error bound, which will be shown in Section 3.4.

### 3.1 IS MSE A GOOD METRIC FOR OFFLINE MBO?

An ideal metric should be able to accurately assess the goodness of a surrogate model, i.e., the better the metric, the better the quality of the final design obtained using the surrogate model. As shown in Eq. (1), $\mathbf{x}_T$, which approximately maximizes the surrogate model $\hat{f}_{\boldsymbol{\theta}}$ by gradient ascent, serves as the final design to output. During the optimization process, it will inevitably traverse the OOD region. Therefore, the performance of the surrogate model in the OOD region will significantly impact the performance of offline MBO. Unfortunately, previous works (Trabucco et al., 2021; 2022) have shown that the regression-based models optimized using MSE often result in poor predictions in the OOD region, i.e., the MSE value in the OOD region (denoted as OOD-MSE) can be very high, and thus many methods have been proposed to decrease OOD-MSE (Fu & Levine, 2021; Chen et al., 2023a; Yuan et al., 2023) or avoid getting into OOD regions (Trabucco et al., 2021; Yu et al., 2021; Yao et al., 2024). In this paper, however, we indicate that even if OOD-MSE is small, the final performance of offline MBO can still be bad. That is, the relationship between the quality of the final designs and OOD-MSE is weak. In the following, we will validate this through experiments.

To analyze the correlation between the OOD-MSE of a surrogate model and the score of the final design candidate obtained by conducting gradient ascent on the surrogate model, we select five surrogate models: a gradient-ascent baseline and four state-of-the-art *forward* approaches, COMs (Trabucco et al., 2021), IOM (Qi et al., 2022), ICT (Yuan et al., 2023), and Tri-Mentoring (Chen et al., 2023a). We follow the default setting as in Chen et al. (2023a); Yuan et al. (2023) for data preparation and model-inner search procedures. To construct an OOD dataset, we follow the approach outlined in Chen et al. (2023a), selecting high-scoring designs that are excluded from the training data in Design-Bench (Trabucco et al., 2022). Detailed information regarding model selection, training and search configurations, and OOD dataset construction can be found in Appendix E.1. We train the surrogate models, evaluate their performance using various metrics (e.g., MSE) on the OOD dataset, and obtain the final design with its corresponding ground-truth score under eight different seeds. Subsequently, we rank the OOD-MSE values in ascending order, and rank the 100th percentile scores of the final designs in descending order. To show the correlation between OOD-MSE and the final score, we create scatter plots of the two rankings and calculate their Spearman correlation coefficient.

The left two subfigures of Figure 2 show the scatter plots on a continuous task, D'Kitty (Ahn et al., 2020), and a discrete task, TF-Bind-8 (Barrera et al., 2016). Both scatter plots exhibit highly dispersed data points, with no clear overall trend or strong clustering, showing no consistent pattern in their distribution. This scattered nature of the data points is also reflected in the low Spearman correlation coefficients ($0.23$ for D'Kitty and $-0.24$ for TF-Bind-8), indicating weak correlations between OOD-MSE rank and score rank in both tasks. These results demonstrate that OOD-MSE is not a good metric for offline MBO, underscoring the need for a more reliable evaluation metric.

### 3.2 WHAT IS THE APPROPRIATE METRIC FOR OFFLINE MBO?

As we mentioned before, an intuition of offline MBO is that the goodness of a surrogate model may depend on its ability to preserve the score ordering of designs dictated by the ground-truth function. We substantiate this intuition through the following theorem.

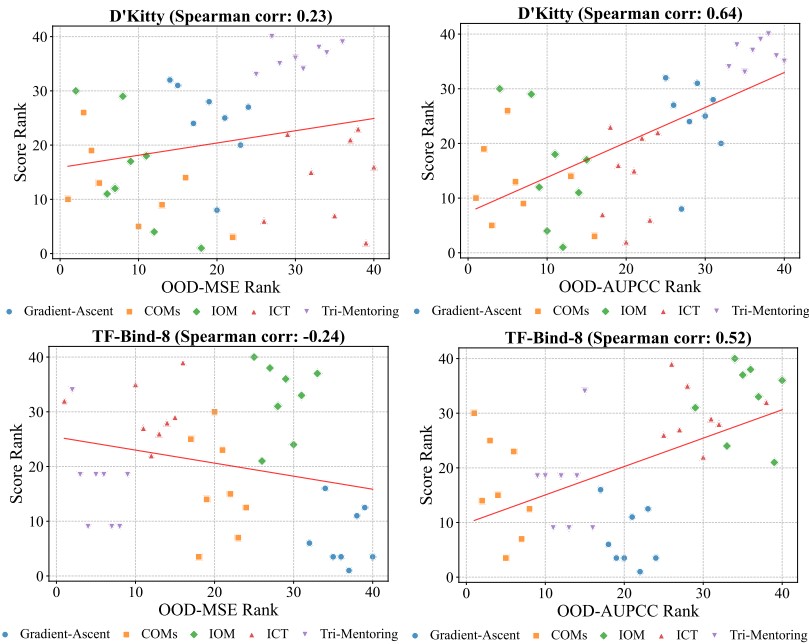

Figure 2: Scatter plots of five surrogate models (each trained using eight seeds) on the two tasks of D'Kitty and TF-Bind-8, where the $y$-axis denotes the rank of the 100th percentile score, and the $x$-axis denotes the rank of the metric in the OOD region, i.e., OOD-MSE or OOD-AUPCC. The Spearman correlation coefficients are also calculated, as shown in the title of each subfigure.

**Theorem 1** (Equivalence of Optima for Order-Preserving Surrogates). *Let $\hat{f}_{\boldsymbol{\theta}}$ be a surrogate model and $f$ the ground-truth function. A function $h : \mathbb{R} \to \mathbb{R}$ is order-preserving, if $\forall y_1, y_2 \in \mathbb{R}$, $y_1 < y_2$ iff $h(y_1) < h(y_2)$. If there exists an order-preserving $h$ such that $\hat{f}_{\boldsymbol{\theta}}(\mathbf{x}) = h(f(\mathbf{x})) \, \forall \mathbf{x}$, then finding the maximum of $f$ is equivalent to finding that of $\hat{f}_{\boldsymbol{\theta}}$, i.e., $\arg\max_{\mathbf{x} \in \mathcal{X}} f(\mathbf{x}) = \arg\max_{\mathbf{x} \in \mathcal{X}} \hat{f}_{\boldsymbol{\theta}}(\mathbf{x})$.*

*Proof.* Suppose $\mathbf{x}^* \in \arg\max_{\mathbf{x}} f(\mathbf{x})$. For any $\mathbf{x}$, we have $f(\mathbf{x}^*) \geq f(\mathbf{x})$. Since $h$ is order-preserving, we have $h(f(\mathbf{x}^*)) \geq h(f(\mathbf{x}))$ for all $\mathbf{x}$. Thus, given $\hat{f}_{\boldsymbol{\theta}}(\mathbf{x}) = h(f(\mathbf{x}))$, we have $\hat{f}_{\boldsymbol{\theta}}(\mathbf{x}^*) \geq \hat{f}_{\boldsymbol{\theta}}(\mathbf{x})$ for all $\mathbf{x}$. Therefore, $\mathbf{x}^* \in \arg\max_{\mathbf{x}} \hat{f}_{\boldsymbol{\theta}}(\mathbf{x})$, i.e., $\arg\max_{\mathbf{x}} f(\mathbf{x}) \subseteq \arg\max_{\mathbf{x}} \hat{f}_{\boldsymbol{\theta}}(\mathbf{x})$. Note that since $h$ is strictly increasing, it is bijective and thus has an inverse function $h^{-1}$, which is also strictly increasing. With $h^{-1}$, the reverse implication follows similarly, proving the equivalence. $\square$

Theorem 1 shows that a good surrogate model needs to maintain an order-preserving mapping from the ground-truth function. Besides, in the practical setting of offline MBO, the standard procedure is to select the top-$k$ designs (e.g., $k = 128$), which maximize the surrogate model's predictions, for evaluation (Trabucco et al., 2022). Thus, we introduce a novel metric, Area Under the Precision-Coverage Curve (AUPCC) in Definition 1, for offline MBO to assess the model's capability in identifying the top-$k$ ones from a set of candidate designs.

**Definition 1** (AUPCC for Offline MBO). *Consider a surrogate model $\hat{f}_{\boldsymbol{\theta}}$ and a ground-truth function $f$. Given a dataset $\mathcal{D}_0 = \{(\mathbf{x}_i, y_i)\}_{i=1}^N$, denote $\{\hat{f}_{\boldsymbol{\theta}}(\mathbf{x}_i)\}_{i=1}^N$ as $\hat{f}_{\boldsymbol{\theta}}(\mathcal{D}_0)$, and $\{f(\mathbf{x}_i)\}_{i=1}^N$ as $f(\mathcal{D}_0)$. Let $\mathrm{top}_k(S)$ denote the set of the $k$ largest elements in set $S$. For each $k \in \{1, 2, ..., N\}$,*

$$\mathrm{Precision}\,@k = \frac{|\,\mathrm{top}_k(\hat{f}_{\boldsymbol{\theta}}(\mathcal{D}_0)) \cap \mathrm{top}_k(f(\mathcal{D}_0))\,|}{|\,\mathrm{top}_k(f(\mathcal{D}_0))\,|} = \frac{|\,\mathrm{top}_k(\hat{f}_{\boldsymbol{\theta}}(\mathcal{D}_0)) \cap \mathrm{top}_k(f(\mathcal{D}_0))\,|}{k},$$

$$\mathrm{Coverage}\,@k = |\,\mathrm{top}_k(\hat{f}_{\boldsymbol{\theta}}(\mathcal{D}_0)) \cap \mathcal{D}_0\,|/|\mathcal{D}_0| = k/N.$$

*The Precision-Coverage curve is obtained by plotting $\mathrm{Precision}\,@k$ against $\mathrm{Coverage}\,@k$ for all values of $k$. Then, the AUPCC is defined as the area under this curve:*

$$\mathrm{AUPCC} \approx \sum_{k=1}^{N-1} (\mathrm{Coverage}\,@(k+1) - \mathrm{Coverage}\,@k) \cdot \frac{\mathrm{Precision}\,@(k+1) + \mathrm{Precision}\,@k}{2}.$$

The AUPCC metric for offline MBO can effectively evaluate a model's ability to identify top-$k$ designs with varying $k$ and thus the ability to preserve order across the entire design space, so it naturally serves as a ranking-related metric. A higher AUPCC value indicates better performance in ranking and selecting better designs. We visualize the correlation between OOD-AUPCC (i.e., the AUPCC value in the OOD region) rank and score rank in the right two subfigures of Figure 2, where the rank of OOD-AUPCC is obtained in descending order. In contrast to the OOD-MSE results, the scatter plots of OOD-AUPCC exhibit clear upward trends, with data points clustered more tightly around the diagonal compared to their OOD-MSE counterparts. This improved correlation is also verified by the substantially higher Spearman correlation coefficients, 0.64 for D'Kitty and 0.52 for TF-Bind-8.

To further validate the reliability of AUPCC compared to MSE, we conduct a quantitative analysis, incorporating another three tasks, Superconductor (Hamidieh, 2018) and Ant (Brockman et al., 2016) in continuous space, and TF-Bind-10 (Barrera et al., 2016) in discrete space. We evaluate the metrics in the OOD regions and the final scores, and calculate Spearman correlation coefficients between the two rankings, following the same approach as in our previous analysis. The results in Table 1 demonstrate the superior performance of OOD-AUPCC compared to OOD-MSE in correlating with the 100th percentile score across various offline MBO tasks. OOD-AUPCC consistently shows stronger correlations than OOD-MSE, with an average improvement of 0.364 in correlation strength. Notably, OOD-AUPCC achieves positive or significantly improved correlations even in the tasks where OOD-MSE shows negative correlations, such as Superconductor and TF-Bind-8 tasks. Coupled with Theorem 1, which establishes the relationship between a model's order-preserving capability and its final performance, the consistently stronger empirical correlations confirm that OOD-AUPCC is indeed a more effective and reliable metric than OOD-MSE for evaluating the performance of a surrogate model in offline MBO. In the next section, we will discuss how to use LTR techniques to optimize the AUPCC, thus to obtain high-scoring designs.

Table 1: Comparison between Spearman correlation coefficients of OOD-MSE and OOD-AUPCC with respect to the 100th percentile score.

| OOD-Metric | Ant | | D'Kitty | | Superconductor | | TF-Bind-8 | | TF-Bind-10 | |
|---|---|---|---|---|---|---|---|---|---|---|
| | Coef. | Gain | Coef. | Gain | Coef. | Gain | Coef. | Gain | Coef. | Gain |
| OOD-MSE | 0.161 | | 0.243 | | -0.116 | | -0.239 | | -0.573 | |
| OOD-AUPCC | 0.257 | **+0.096** | 0.503 | **+0.260** | 0.101 | **+0.217** | 0.520 | **+0.759** | -0.087 | **+0.486** |

### 3.3 OFFLINE MBO BY LEARNING TO RANK: A PRACTICAL ALGORITHM

In this section, in order to optimize AUPCC for the surrogate model, we design a novel framework for offline MBO based on LTR, as shown in Algorithm 1, which consists three parts: 1) *data augmentation*; 2) *LTR loss learning*; 3) *output adaptation*.

**Data augmentation.** In LTR tasks, the training set $\mathcal{D}_{\mathrm{R}}$ typically requires a list of designs as features. However, the offline dataset $\mathcal{D}$ in offline MBO is not directly structured in this manner, thus the LTR loss functions cannot be directly applied. A naïve approach to address this issue is to treat each batch of training data as a list of designs to be ranked, with the batch size determining the list length. However, this method has its limitation since each design in the training data appears in only one list during one single epoch, which is unable to analyze its relationship with other designs that are not in the list. To address this limitation, we propose a simple yet effective data augmentation method. We randomly sample $m$ design-score pairs $\{(\mathbf{x}_i, y_i)\}_{i=1}^m$ from $\mathcal{D}$, and concatenate them to form a design list $\mathbf{X} = [\mathbf{x}_1, \mathbf{x}_2, \ldots, \mathbf{x}_m]^\top$ and its score list $\mathbf{y} = [y_1, y_2, \ldots, y_m]^\top$; then repeat this step for $n$ times to construct a dataset $\mathcal{D}_{\mathrm{R}} = \{(\mathbf{X}_i, \mathbf{y}_i)\}_{i=1}^n$ for LTR modeling. We will discuss the setting of $n$ and $m$ in Section 4.1, and show the benefit of data augmentation over the naïve approach in Section 4.2.

**LTR loss learning.** In Section 3.2, we have discussed that AUPCC is a ranking-related metric, and thus we can use the well-studied ranking loss (Li, 2011) from the field of LTR to optimize the AUPCC on the training distribution, so as to generalize to the OOD regions. We study a wide range of ranking losses, including pointwise (Crammer & Singer, 2001), pairwise (Köppel et al., 2019), and listwise (Xia et al., 2008) losses. Here we take RankCosine (Qin et al., 2008), a pairwise loss, and ListNet (Cao et al., 2007), a listwise loss, for example. The idea of RankCosine is to measure the difference between predicted and true rankings using cosine similarity, operating directly in the

---

**Algorithm 1** Offline MBO by Learning to Rank

---

**Input**: Offline dataset $\mathcal{D}$, number $n$ of lists in the training data, length $m$ of each list, training steps $N_0$, ranking loss $l$, learning rate $\lambda$, search steps $T$, search step size $\eta$.
**Output**: The final high-scoring design candidate.

1: Initialize $\hat{f}_{\boldsymbol{\theta}}$; Initialize $\mathbf{x}_0$ as the design with the highest score in $\mathcal{D}$;
2: Initialize $\mathcal{D}_{\mathrm{R}} \leftarrow \emptyset$;                                      ▷ *Construct training data via data augmentation*
3: **for** $i = 1$ to $n$ **do**
4:     Randomly sample $m$ design-score pairs $(\mathbf{x}, y)$ from $\mathcal{D}$;
5:     Add $(\mathbf{X}, \mathbf{y})$ to $\mathcal{D}_{\mathrm{R}}$, where $\mathbf{X} = [\mathbf{x}_1, \mathbf{x}_2, \ldots, \mathbf{x}_m]^\top$ and $\mathbf{y} = [y_1, y_2, \ldots, y_m]^\top$
6: **for** $i = 1$ to $N_0$ **do**                                      ▷ *Use LTR loss to train the surrogate model*
7:     Calculate the ranking loss: $\mathcal{L}(\boldsymbol{\theta}) = \frac{1}{|\mathcal{D}_{\mathrm{R}}|} \sum_{(\mathbf{X}, \mathbf{y}) \in \mathcal{D}_{\mathrm{R}}} l(\mathbf{y}, \hat{f}_{\boldsymbol{\theta}}(\mathbf{X}))$,
        where $\hat{f}_{\boldsymbol{\theta}}(\mathbf{X}) = [\hat{f}_{\boldsymbol{\theta}}(\mathbf{x}_1), \hat{f}_{\boldsymbol{\theta}}(\mathbf{x}_2), \ldots, \hat{f}_{\boldsymbol{\theta}}(\mathbf{x}_m)]^\top$;
8:     Minimize $\mathcal{L}(\boldsymbol{\theta})$ with respect to $\boldsymbol{\theta}$ using gradient update: $\boldsymbol{\theta} \leftarrow \boldsymbol{\theta} - \lambda \nabla_{\boldsymbol{\theta}} \mathcal{L}(\boldsymbol{\theta})$
9: Calculate the in-distribution predictions $\tilde{\mathbf{y}} = \{\tilde{y} \mid \tilde{y} = \hat{f}_{\boldsymbol{\theta}}(\mathbf{x}), (\mathbf{x}, y) \in \mathcal{D}\}$;
                                                     ▷ *Conduct gradient ascent via output adaptation*
10: Obtain statistics of the in-distribution predictions: $\tilde{\mu} = \mathrm{mean}(\tilde{\mathbf{y}})$, $\tilde{\sigma} = \mathrm{std}(\tilde{\mathbf{y}})$;
11: **for** $t = 0$ to $T - 1$ **do**
12:     Update $\mathbf{x}_{t+1}$ via gradient ascent: $\mathbf{x}_{t+1} = \mathbf{x}_t + \eta \nabla_{\mathbf{x}} \mathcal{L}_{\mathrm{opt}}(\mathbf{x})|_{\mathbf{x} = \mathbf{x}_t}$,
        where $\mathcal{L}_{\mathrm{opt}}(\mathbf{x}) := (\hat{f}_{\boldsymbol{\theta}}(\mathbf{x}) - \tilde{\mu})/\tilde{\sigma}$
13: Return $\mathbf{x}_T$

---

score space. Formally, given a list $\mathbf{X}$ of designs and the list $\mathbf{y}$ of their corresponding scores, let $\hat{f}_{\boldsymbol{\theta}}(\mathbf{X}) = [\hat{f}_{\boldsymbol{\theta}}(\mathbf{x}_1), \hat{f}_{\boldsymbol{\theta}}(\mathbf{x}_2), \ldots, \hat{f}_{\boldsymbol{\theta}}(\mathbf{x}_m)]^\top$ be the predicted scores. The RankCosine loss function is:

$$l_{RankCosine}(\mathbf{y}, \hat{f}_{\boldsymbol{\theta}}(\mathbf{X})) = 1 - \mathbf{y} \cdot \hat{f}_{\boldsymbol{\theta}}(\mathbf{X})/(\|\mathbf{y}\| \cdot \|\hat{f}_{\boldsymbol{\theta}}(\mathbf{X})\|).$$

The idea of ListNet is to minimize the cross-entropy between the predicted ranking distribution and the true ranking distribution, which is defined as:

$$l_{ListNet}(\mathbf{y}, \hat{f}_{\boldsymbol{\theta}}(\mathbf{X})) = -\sum_{j=1}^{m} \frac{\exp(y_j)}{\sum_{i=1}^{m} \exp(y_i)} \log \frac{\exp(\hat{f}_{\boldsymbol{\theta}}(\mathbf{x}_j))}{\sum_{i=1}^{m} \exp(\hat{f}_{\boldsymbol{\theta}}(\mathbf{x}_i))}.$$

We provide detailed description of other ranking losses in Appendix D, and compare their effectiveness for offline MBO in Section 4.2. We also provide detailed information of model training in Section 4.1.

**Output adaptation.** The surrogate model trained with ranking loss has a crucial issue for the hyper-parameter setting of gradient-ascent optimizers. Unlike MSE, which aims for accurate prediction of target scores, ranking losses do not require precise estimation of target scores. This shift in objective may lead to significant changes in the scale of model predictions, and thus impact the magnitude of gradients, making it challenging to determine appropriate values for the search step size $\eta$ and the number $T$ of search steps in Eq. (1). Moreover, different ranking losses can result in different output scales, which necessitate careful hyper-parameter tuning for a specific loss.

Notably, the scores in the training data for regression-based models have a statistical characteristic of zero mean and unit standard deviation after z-score normalization (Trabucco et al., 2021; 2022), and the trained regression-based model will try to preserve these statistical properties within the training distribution. Consequently, to mitigate the impact of varying scales across different loss functions and to ensure a fair comparison with the regression-based models, we normalize the predictions of the ranking model after it is trained. Specifically, we first apply the trained model to the entire training set and calculate the mean value $\tilde{\mu}$ and standard deviation $\tilde{\sigma}$ of the resulting predictions. Subsequently, we use $\tilde{\mu}$ and $\tilde{\sigma}$ to apply z-score normalization to the model's prediction. Such normalization enables us to directly use the setting of $\eta$ and $T$ as in regression-based models. That is, we compute the gradient of the normalized predictions with respect to $\mathbf{x}$, and use the default hyper-parameters in Chen et al. (2023a); Yuan et al. (2023) to search for the final design candidate. We will examine the effectiveness of using output adaptation in Section 4.2.

### 3.4 THEORETICAL ANALYSIS

In the previous subsections, we have indicated the importance of preserving the score order of designs, and proposed to learn a surrogate model by optimizing ranking losses. Here, we further point out that the generalization error can be well bounded in the context of LTR. Note that the generalization of LTR has been well studied (Agarwal et al., 2005; Lan et al., 2009; Chen et al., 2010; Tewari & Chaudhuri, 2015), which is mainly analyzed by the Probably Approximately Correct (PAC) learning theory (Cucker & Smale, 2001) and Rademacher Complexity (Bartlett & Mendelson, 2003). By leveraging these existing generalization error bounds, we provide theoretical support for our approach of applying LTR techniques for offline MBO.

Formally, assume that we have an i.i.d. training data $\mathcal{D}_{\mathrm{R}} = \{(\mathbf{X}_i, \mathbf{y}_i)\}_{i=1}^n$ where $\mathbf{X}_i \in \mathcal{X}^m$, consisting of $m$ designs, and $\mathbf{y}_i \in \mathbb{R}^m$. Given a ranking algorithm $\mathcal{A}$ (e.g., RankCosine or ListNet), its loss function $l_{\mathcal{A}}(f; \mathbf{X}, \mathbf{y})$ is normalized by $l_{\mathcal{A}}(f; \mathbf{X}, \mathbf{y})/Z_{\mathcal{A}}$, where $Z_{\mathcal{A}}$ is a normalization constant (e.g., $Z_{RankCosine} = 1$). The *expected risk* with respect to the algorithm $\mathcal{A}$ is defined as $R_{l_{\mathcal{A}}}(f) = \int_{\mathcal{X}^m \times \mathbb{R}^m} l_{\mathcal{A}}(f; \mathbf{X}, \mathbf{y}) P(\mathrm{d}\mathbf{X}, \mathrm{d}\mathbf{y})$, and the *empirical risk* is defined as $\hat{R}_{l_{\mathcal{A}}}(f; \mathcal{D}_{\mathrm{R}}) = \frac{1}{n} \sum_{i=1}^n l_{\mathcal{A}}(f; \mathbf{X}_i, \mathbf{y}_i)$. Let $\mathcal{F}$ be the ranking function class, and Theorem 2 gives an upper bound on the generalization error $\sup_{f \in \mathcal{F}}(R_{l_{\mathcal{A}}}(f) - \hat{R}_{l_{\mathcal{A}}}(f; \mathcal{D}_{\mathrm{R}}))$.

**Theorem 2** (Generalization Error Bound for LTR (Lan et al., 2009))**.** *Let $\phi$ be an increasing and strictly positive transformation function (e.g., $\phi(z) = \exp(z)$). Assume that: 1) $\forall \mathbf{x} \in \mathcal{X}$, $\|\mathbf{x}\| \leq M$; 2) the ranking model $f$ to be learned is from the linear function class $\mathcal{F} = \{\mathbf{x} \to \mathbf{w}^\top \mathbf{x} \mid \|\mathbf{w}\| \leq B\}$. Then with probability $1 - \delta$, the following inequality holds:*

$$\sup_{f \in \mathcal{F}} \left( R_{l_{\mathcal{A}}}(f) - \hat{R}_{l_{\mathcal{A}}}(f; \mathcal{D}_R) \right) \leq 4BM \cdot C_{\mathcal{A}}(\phi) N(\phi)/\sqrt{n} + \sqrt{2 \ln (2/\delta)/n},$$

*where: 1) $\mathcal{A}$ stands for a specific LTR algorithm; 2) $N(\phi) = \sup_{z \in [-BM, BM]} \phi'(z)$, which is an algorithm-independent factor measuring the smoothness of $\phi$; 3) $C_{\mathcal{A}}(\phi)$ is an algorithm-dependent factor, e.g., $C_{RankCosine}(\phi) = \sqrt{m}/(2\phi(-BM))$.*

We will introduce some settings of $\phi$ and the corresponding $N(\phi)$ and $C_{\mathcal{A}}(\phi)$ in Appendix B. We can observe from the inequality in Theorem 2 that the generalization error bound vanishes at the rate $\mathcal{O}(1/\sqrt{n})$, since $C_{\mathcal{A}}(\phi)$ and $N(\phi)$ are independent of the size $n$ of training set. In Appendix C, we discuss probable approaches and difficulties in extending the theoretical analysis, identify a special case where the pairwise ranking loss is more robust than MSE, and analyze it via experiments.

## 4 EXPERIMENTS

In this section, we empirically compare the proposed method with a large variety of previous offline MBO methods on various tasks. First, we introduce our experimental settings, including five tasks, twenty compared methods, training settings, and evaluation metrics. Then, we present the results to show the superiority of our method. We also examine the influence of using different ranking losses, and conduct ablation studies to investigate the effectiveness of each module of our method. Furthermore, we simply replace MSE of existing methods with the best-performing ranking loss, to demonstrate the versatility of the ranking loss for offline MBO. Finally, we provide the metrics, OOD-MSE and OOD-AUPCC, in the OOD regions to validate their relationship with the final performance. Our implementation is available at `https://github.com/lamda-bbo/Offline-RaM`.

### 4.1 EXPERIMENTAL SETTINGS

**Benchmark and tasks.** We benchmark our method on Design-Bench tasks (Trabucco et al., 2022), including three continuous tasks and two discrete tasks [1]. The continuous tasks include: 1) **Ant Morphology** (Brockman et al., 2016): identify an ant morphology with 60 parameters to crawl quickly. 2) **D'Kitty Morphology** (Ahn et al., 2020): optimize a D'Kitty morphology with 56 parameters to crawl quickly. 3) **Superconductor** (Hamidieh, 2018): design a 86-dimensional superconducting material to maximize the critical temperature. The two discrete tasks are **TF-Bind-8** and **TF-Bind-10** (Barrera et al., 2016): find a DNA sequence of length 8 and 10, respectively, maximizing binding affinity with a particular transcription factor.

---

[1]Following recent works (Yun et al., 2024; Yu et al., 2024), we exclude three tasks from Design-Bench, and provide detailed explanations in Appendix E.2.

**Compared methods.** We mainly consider three categories of methods to solve offline MBO. The first category involves baselines that optimize a trained regression-based model, such as BO-$q$EI (Garnett, 2023; Shahriari et al., 2016), CMA-ES (Hansen, 2016), REINFORCE (Williams, 1992), Gradient Ascent and its variants of mean ensemble and min ensemble. The second category encompasses *backward* approaches, including CbAS (Brookes et al., 2019), MINs (Kumar & Levine, 2020), DDOM (Krishnamoorthy et al., 2023), BONET (Mashkaria et al., 2023), and GTG (Yun et al., 2024). The third category comprises recently proposed *forward* approaches, which contain COMs (Trabucco et al., 2021), RoMA (Yu et al., 2021), IOM (Qi et al., 2022), BDI (Chen et al., 2022), ICT (Yuan et al., 2023), Tri-Mentoring (Chen et al., 2023a), PGS (Chemingui et al., 2024), FGM (Kuba et al., 2024b), and Match-OPT (Hoang et al., 2024) [2].

**Training settings.** We set the size $n$ of training dataset to $10,000$, and following LETOR 4.0 (Qin & Liu, 2013; Qin et al., 2010b), a prevalent benchmark for LTR, we set the list length $m = 1000$. To make a fair comparison to regression-based methods, following Trabucco et al. (2021; 2022); Chen et al. (2023a); Yuan et al. (2023), we model the surrogate model $\hat{f}_{\boldsymbol{\theta}}$ as a simple multilayer perceptron with two hidden layers of size 2048 using PyTorch (Paszke et al., 2019). We use ReLU as activation functions. RankCosine (Qin et al., 2008) and ListNet (Cao et al., 2007) will be used as two main loss functions in our experiments. The model is optimized using Adam (Kingma & Ba, 2015) with a learning rate of $3 \times 10^{-4}$ and a weight decay coefficient of $1 \times 10^{-5}$. After the model is trained, following Chen et al. (2023a); Yuan et al. (2023), we set $\eta = 1 \times 10^{-3}$ and $T = 200$ for continuous tasks, and $\eta = 1 \times 10^{-1}$ and $T = 100$ for discrete tasks to search for the final design. All experiments are conducted using eight different seeds. Additional training details are provided in Appendix E.4.

**Evaluation and metrics.** For evaluation, we use the oracle from Design-Bench and follow the protocol of prior works (Trabucco et al., 2021; 2022). That is, we identify $k = 128$ most promising designs selected by an algorithm and report the 100th percentile normalized ground-truth score. A design score $y$ is normalized via computing $(y - y_{\min})/(y_{\min} - y_{\max})$, where $y_{\min}$ and $y_{\max}$ denote the lowest and the highest scores in the full unobserved dataset from Design-Bench. We also provide the 50th percentile normalized ground-truth results in Appendix F.1.

## 4.2 EXPERIMENTAL RESULTS

**Main results.** In Table 2, we report the results of our experiments, where our method based on **Ra**nking **M**odel is denoted as **RaM** appended with the name of the employed ranking loss. Among the compared 22 methods, RaM-RankCosine and RaM-ListNet achieve the two best average ranks, 2.7 and 2.2, respectively, while the third best method, BDI, only obtains an average rank of 5.9. We can observe that RaM-RankCosine performs best on one task, TF-Bind-10, and is runner-up on two tasks, Superconductor and TF-Bind-8; and RaM-ListNet performs best on two tasks, D'Kitty and Superconductor. These results clearly demonstrate the superior performance of our proposed method.

**Influence of different ranking loss.** We compare RaM with various ranking losses: Sigmoid-CrossEntropy (SCE), BinaryCrossEntropy (BCE), and MSE [3] for pointwise loss; RankNet (Burges et al., 2005), LambdaRank (Burges et al., 2006; Wang et al., 2018), and RankCosine (Qin et al., 2008) for pairwise loss; Softmax (Cao et al., 2007; Bruch et al., 2019a), ListNet (Cao et al., 2007), ListMLE (Xia et al., 2008), and ApproxNDCG (Qin et al., 2010a; Bruch et al., 2019b) for listwise loss. The results in Table 8 in Appendix F.2 show that ListNet is the best-performing loss with an average rank of 2.0 over 10 losses, and RankCosine is the runner-up with an average rank of 3.2.

**Ablation of main modules.** To better validate the effectiveness of the two moduels, *data augmentation* and *output adaptation*, of our method, we perform ablation studies based on the top-performing loss functions shown in Table 8: MSE for pointwise loss, RankCosine for pairwise loss, and ListNet for listwise loss. The results in Table 9 in Appendix F.3 show that for each considered loss, RaM with data augmentation performs better than the naïve approach which treats a batch of the dataset as a list to rank. The results in Table 10 show the benefit of using output adaptation. We also examine the influence of the list length $m$, as illustrated in Appendix F.4.

---

[2]Due to the lack of open-source implementations or inapplicability for comparison, we exclude NEMO (Fu & Levine, 2021), BOSS (Dao et al., 2024b), DEMO (Yuan et al., 2024) and LEO (Yu et al., 2024). Detailed explanations are provided in Appendix E.3.

[3]Note that MSE is a regression loss, which thus can be viewed as a pointwise ranking loss.

Table 2: 100th percentile normalized score in Design-Bench, where the best and runner-up results on each task are **Blue** and **Violet**. $\mathcal{D}$(best) denotes the best score in the offline dataset.

| Method | Ant | D'Kitty | Superconductor | TF-Bind-8 | TF-Bind-10 | Mean Rank |
|---|---|---|---|---|---|---|
| $\mathcal{D}$(best) | 0.565 | 0.884 | 0.400 | 0.439 | 0.467 | / |
| BO-$q$EI | 0.812 ± 0.000 | 0.896 ± 0.000 | 0.382 ± 0.013 | 0.802 ± 0.081 | 0.628 ± 0.036 | 18.0 / 22 |
| CMA-ES | **1.712 ± 0.754** | 0.725 ± 0.002 | 0.463 ± 0.042 | 0.944 ± 0.017 | 0.641 ± 0.036 | 11.4 / 22 |
| REINFORCE | 0.248 ± 0.039 | 0.541 ± 0.196 | 0.478 ± 0.017 | 0.935 ± 0.049 | **0.673 ± 0.074** | 14.0 / 22 |
| Grad. Ascent | 0.273 ± 0.023 | 0.853 ± 0.018 | 0.510 ± 0.028 | 0.969 ± 0.021 | 0.646 ± 0.037 | 11.6 / 22 |
| Grad. Ascent Mean | 0.306 ± 0.053 | 0.875 ± 0.024 | 0.508 ± 0.019 | **0.985 ± 0.008** | 0.633 ± 0.030 | 11.2 / 22 |
| Grad. Ascent Min | 0.282 ± 0.033 | 0.884 ± 0.018 | **0.514 ± 0.020** | 0.979 ± 0.014 | 0.632 ± 0.027 | 11.5 / 22 |
| CbAS | 0.846 ± 0.032 | 0.896 ± 0.009 | 0.421 ± 0.049 | 0.921 ± 0.046 | 0.630 ± 0.039 | 15.5 / 22 |
| MINs | 0.906 ± 0.024 | 0.939 ± 0.007 | 0.464 ± 0.023 | 0.910 ± 0.051 | 0.633 ± 0.034 | 13.0 / 22 |
| DDOM | 0.908 ± 0.024 | 0.930 ± 0.005 | 0.452 ± 0.028 | 0.913 ± 0.047 | 0.616 ± 0.018 | 14.6 / 22 |
| BONET | 0.921 ± 0.031 | 0.949 ± 0.016 | 0.390 ± 0.022 | 0.798 ± 0.123 | 0.575 ± 0.039 | 15.1 / 22 |
| GTG | 0.855 ± 0.044 | 0.942 ± 0.017 | 0.480 ± 0.055 | 0.910 ± 0.040 | 0.619 ± 0.029 | 13.9 / 22 |
| COMs | 0.916 ± 0.026 | 0.949 ± 0.016 | 0.460 ± 0.040 | 0.953 ± 0.038 | 0.644 ± 0.052 | 9.5 / 22 |
| RoMA | 0.430 ± 0.048 | 0.767 ± 0.031 | 0.494 ± 0.025 | 0.665 ± 0.000 | 0.553 ± 0.000 | 18.3 / 22 |
| IOM | 0.889 ± 0.034 | 0.928 ± 0.008 | 0.491 ± 0.034 | 0.925 ± 0.054 | 0.628 ± 0.036 | 13.1 / 22 |
| BDI | **0.963 ± 0.000** | 0.941 ± 0.000 | 0.508 ± 0.013 | 0.973 ± 0.000 | 0.658 ± 0.000 | 5.9 / 22 |
| ICT | 0.915 ± 0.024 | 0.947 ± 0.009 | 0.494 ± 0.026 | 0.897 ± 0.050 | 0.659 ± 0.024 | 9.4 / 22 |
| Tri-Mentoring | 0.891 ± 0.011 | 0.947 ± 0.005 | 0.503 ± 0.013 | 0.956 ± 0.000 | 0.662 ± 0.012 | 7.7 / 22 |
| PGS | 0.715 ± 0.046 | **0.954 ± 0.022** | 0.444 ± 0.020 | 0.889 ± 0.061 | 0.634 ± 0.040 | 13.2 / 22 |
| FGM | 0.923 ± 0.023 | 0.944 ± 0.014 | 0.481 ± 0.024 | 0.811 ± 0.079 | 0.611 ± 0.008 | 13.2 / 22 |
| Match-OPT | 0.933 ± 0.016 | 0.952 ± 0.008 | 0.504 ± 0.021 | 0.824 ± 0.067 | 0.655 ± 0.050 | 8.0 / 22 |
| **RaM-RankCosine (Ours)** | 0.940 ± 0.028 | 0.951 ± 0.017 | **0.514 ± 0.026** | **0.982 ± 0.012** | **0.675 ± 0.049** | **2.7 / 22** |
| **RaM-ListNet (Ours)** | 0.949 ± 0.025 | **0.962 ± 0.015** | **0.517 ± 0.029** | 0.981 ± 0.012 | 0.670 ± 0.035 | **2.2 / 22** |

Table 3: 100th percentile normalized score of different methods combined with the MSE or ListNet loss in Design-Bench, where positive and negative gain rates are **Blue** and **Red**.

| Method | Type | Ant | | D'Kitty | | Superconductor | | TF-Bind-8 | | TF-Bind-10 | |
|---|---|---|---|---|---|---|---|---|---|---|---|
| | | Score | Gain | Score | Gain | Score | Gain | Score | Gain | Score | Gain |
| BO-$q$EI | MSE | 0.812 ± 0.000 | | 0.896 ± 0.000 | | 0.382 ± 0.013 | | 0.802 ± 0.081 | | 0.628 ± 0.036 | |
| | ListNet | 0.812 ± 0.000 | +0.0% | 0.896 ± 0.000 | +0.0% | 0.509 ± 0.013 | +33.2% | 0.912 ± 0.032 | +13.7% | 0.653 ± 0.056 | +4.0% |
| CMA-ES | MSE | 1.712 ± 0.705 | | 0.722 ± 0.001 | | 0.463 ± 0.042 | | 0.944 ± 0.017 | | 0.641 ± 0.036 | |
| | ListNet | 1.923 ± 0.773 | +12.3% | 0.723 ± 0.002 | +0.1% | 0.486 ± 0.020 | +5.0% | 0.960 ± 0.008 | +1.7% | 0.661 ± 0.044 | +3.1% |
| REINFORCE | MSE | 0.248 ± 0.039 | | 0.344 ± 0.091 | | 0.478 ± 0.017 | | 0.935 ± 0.049 | | 0.673 ± 0.074 | |
| | ListNet | 0.318 ± 0.056 | +28.2% | 0.359 ± 0.139 | +4.3% | 0.501 ± 0.013 | +4.8% | 0.935 ± 0.049 | +0.0% | 0.673 ± 0.074 | +0.0% |
| Grad. Ascent | MSE | 0.273 ± 0.022 | | 0.853 ± 0.017 | | 0.510 ± 0.028 | | 0.969 ± 0.020 | | 0.646 ± 0.037 | |
| | ListNet | 0.280 ± 0.021 | +2.6% | 0.890 ± 0.019 | +4.3% | 0.521 ± 0.012 | +2.0% | 0.985 ± 0.011 | +1.7% | 0.660 ± 0.049 | +2.2% |
| CbAS | MSE | 0.846 ± 0.030 | | 0.896 ± 0.009 | | 0.421 ± 0.046 | | 0.921 ± 0.046 | | 0.630 ± 0.039 | |
| | ListNet | 0.854 ± 0.037 | +0.9% | 0.898 ± 0.009 | +0.2% | 0.425 ± 0.036 | +1.0% | 0.956 ± 0.033 | +3.8% | 0.642 ± 0.034 | +1.9% |
| MINs | MSE | 0.906 ± 0.024 | | 0.939 ± 0.007 | | 0.464 ± 0.023 | | 0.910 ± 0.051 | | 0.633 ± 0.032 | |
| | ListNet | 0.911 ± 0.025 | +0.5% | 0.941 ± 0.009 | +0.2% | 0.477 ± 0.019 | +2.8% | 0.910 ± 0.029 | +0.0% | 0.638 ± 0.037 | +0.8% |
| Tri-Mentoring | MSE | 0.891 ± 0.011 | | 0.947 ± 0.005 | | 0.503 ± 0.013 | | 0.956 ± 0.000 | | 0.662 ± 0.012 | |
| | ListNet | 0.915 ± 0.024 | +2.7% | 0.943 ± 0.004 | -0.4% | 0.503 ± 0.010 | +0.0% | 0.971 ± 0.005 | +1.7% | 0.710 ± 0.020 | +7.3% |
| PGS | MSE | 0.715 ± 0.046 | | 0.954 ± 0.022 | | 0.444 ± 0.020 | | 0.889 ± 0.061 | | 0.634 ± 0.040 | |
| | ListNet | 0.723 ± 0.032 | +1.1% | 0.962 ± 0.018 | +0.8% | 0.452 ± 0.042 | +1.8% | 0.886 ± 0.003 | -0.3% | 0.643 ± 0.030 | +1.4% |
| Match-OPT | MSE | 0.933 ± 0.016 | | 0.952 ± 0.008 | | 0.504 ± 0.021 | | 0.824 ± 0.067 | | 0.655 ± 0.050 | |
| | ListNet | 0.936 ± 0.027 | +0.3% | 0.956 ± 0.018 | +0.4% | 0.513 ± 0.011 | +1.8% | 0.829 ± 0.009 | +0.6% | 0.659 ± 0.037 | +0.6% |

**Versatility of ranking loss.** We examine whether simply replacing the MSE loss of some regression-based methods with a ranking loss can even bring improvement. Specifically, we substitute MSE with the best-performing ranking loss, ListNet, and incorporate output adaptation. The results in Table 3 show that the gains are always positive except two cases, clearly demonstrating the versatility of ranking loss. Details regarding method selection and implementation are provided in Appendix E.5.

**Results on OOD-MSE and OOD-AUPCC.** We also present the OOD-MSE and OOD-AUPCC values of some methods in Appendix F.5, where RaM performs well in OOD-AUPCC while poor in OOD-MSE, further demonstrating that ranking loss is more suitable than MSE for offline MBO.

## 5 CONCLUSION

Offline MBO methods often learn a surrogate model by minimizing MSE. In this paper, we question this practice. We empirically show that MSE has a low correlation with the final performance of the surrogate model. Instead, we show that the ranking-related metric AUPCC is well-aligned with the primary goal of offline MBO, and propose a ranking-based model for offline MBO. Extensive experimental results show the superiority of our proposed ranking-based model over a large variety of state-of-the-art offline MBO methods. We hope this work can open a new line of offline MBO.

ACKNOWLEDGMENTS

The authors would like to thank Yi-Xiao He for insightful discussions. This work was supported by the National Science and Technology Major Project (2022ZD0116600), the National Science Foundation of China (62276124, 624B1025, 624B2069), the Fundamental Research Funds for the Central Universities (14380020), and Young Elite Scientists Sponsorship Program by CAST for PhD Students. Shen-Huan Lyu was supported by the National Natural Science Foundation of China (62306104), Hong Kong Scholars Program (XJ2024010), Jiangsu Science Foundation (BK20230949), China Postdoctoral Science Foundation (2023TQ0104), and Jiangsu Excellent Postdoctoral Program (2023ZB140). The authors want to acknowledge support from the Huawei Technology cooperation Project.

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

# A    RELATED WORK

## A.1    OFFLINE MODEL-BASED OPTIMIZATION

Offline MBO methods (Trabucco et al., 2022; Kim et al., 2025; Qian et al., 2025; Xue et al., 2024) can be generally categorized into two types of approaches. The mainstream approach for offline MBO is the *forward* approach, which first trains a forward surrogate model $\hat{f}_{\boldsymbol{\theta}} : \mathcal{X} \rightarrow \mathbb{R}$ and then employs gradient ascent to optimize the learned surrogate to output candidate solutions, as introduced in Section 2. A crucial challenge of this approach is how to improve the surrogate model's generalization ability in the OOD regions, which can significantly affect the performance. Prior works of forward approach mainly add regularization items to: 1) **regulate the nature of the surrogate model**: NEMO (Fu & Levine, 2021) optimizes the gap between the surrogate model and the ground-truth function via normalized maximum likelihood, BOSS (Dao et al., 2024b) and IGNITE (Dao et al., 2024a) regulate the sensitivity of the surrogate model against perturbation on model weights from different perspectives, while RoMA (Yu et al., 2021) enhances the smoothness of the model in a pre-trained and adaptation manner; 2) **regulate surrogate model's predictions directly**: COMs (Trabucco et al., 2021) penalize identified outliers via a GAN-like procedure (Goodfellow et al., 2014), whereas IOM (Qi et al., 2022) maintains representation invariance between the training dataset and design candidate. Given that an ensemble of surrogate models can bring an improvement (Trabucco et al., 2022), ICT (Yuan et al., 2023) and Tri-Mentoring (Chen et al., 2023a) train three symmetric surrogate models and ensemble them, where ICT uses a semi-supervised learning via pseudo-label procedure (Verma et al., 2022) and Tri-Mentoring employs a strategy similar to Tri-training (Zhou & Li, 2005) from a pairwise perspective. Besides, recent works have tried to **uncovering the structural information of the dataset** for better learning. BDI (Chen et al., 2022) utilizes both forward and backward mappings to distill knowledge from the offline dataset to the design. Both FGM (Kuba et al., 2024b) and Cliqueformer (Kuba et al., 2024a) consider a novel modeling, which splits the design space into cliques on dimension-level, to approximate scores. Both PGS (Chemingui et al., 2024) and Match-OPT (Hoang et al., 2024) construct trajectories from the dataset, while PGS uses offline reinforcement learning to learn a policy that predicts the search step size of the gradient ascent optimizer and Match-OPT enforces the model to match the ground-truth gradient. Recent works also consider **regulating the model-inner search procedure**. For example, DEMO (Yuan et al., 2024) edits the designs obtained by gradient ascent via a diffusion prior, GAMBO (Yao et al., 2024) regulates the optimize trajectory via modeling the search procedure as a constrained optimization problem, while ARCOO (Lu et al., 2023) guides the search step size via a trained energy model. However, all prior works in forward approach train the surrogate model *based on a regression-based model* using MSE as a base term in the loss function. In this work, we challenge this practice and train the surrogate model in a ranking suite, obtaining superior performance, as shown in Section 4.

Another type of approach for offline MBO is the *backward* approach, which typically involves training a conditioned generative model $p_{\boldsymbol{\theta}}(\mathbf{x}|y)$ and sampling from it conditioned on a high score, for example, MINs (Kumar & Levine, 2020) trains an inverse mapping using a conditioned GAN-like model (Goodfellow et al., 2014; Mirza & Osindero, 2014), while CbAS (Brookes et al., 2019; Fannjiang & Listgarten, 2020) models it as a zero-sum game via a VAE (Kingma & Welling, 2014). Note that generative models show impressive expressiveness and have achieved huge success, works in this field employ powerful generative model to obtain final designs. DDOM (Krishnamoorthy et al., 2023) and RGD (Chen et al., 2024) directly parameterize the inverse mapping with a conditional diffusion model (Ho et al., 2020) in the design space. ExPT (Nguyen et al., 2023) learns from synthetic prior and adapt in a few-shot suite using Transformers (Vaswani et al., 2017). LEO (Yu et al., 2024) constructs a latent space through an energy-based model that does not require MCMC sampling. Recent works in this category also focus on generating designs via constructed trajectories. For example, BONET (Mashkaria et al., 2023) uses trajectories to mimic a black-box optimizer, thus to train an autoregressive model and sample designs using a heuristic; GTG (Yun et al., 2024) considers improving the quality of trajectories via local search, and then directly generate trajectories using a context conditioning diffusion model.

In the field of offline MBO, some studies are related to the idea of ranking designs or implicitly use the ranking information: 1) Match-OPT (Hoang et al., 2024). The idea of gradient matching in this paper is related to ranking samples, since a model with proper gradient could reflect the relationship in a small neighborhood. 2) Tri-Mentoring (Chen et al., 2023a). In Tri-Mentoring, each proxy uses weak semi-supervised pairwise-ranking-based voting signals provided by other proxies to fix its

predictions and finetune its weights. 3) BONET (Mashkaria et al., 2023). The trajectories used in BONET are constructed by ranking the collected samples, from which the model may capture some ranking information. Although these methods capture ranking information in some ways, in this work, we explicitly identify the idea of ranking samples, and conduct a systematic analysis on this view. After that, we reformulate the objective of the training process by replacing the core MSE loss with a ranking loss, and apply data augmentation and output adaptation for model training and solution search, respectively. The superior experimental results in Section 4 also indicate the significance to focus on ranking information in offline MBO. In Appendix A.2, we also discuss some related works that leverage LTR techniques into their respective fields to make advances.

### A.2 LEVERAGING LTR TECHNIQUES INTO SPECIFIC DOMAINS

In this subsection, we also briefly introduce some works in three fields that share a similar motivation to leverage LTR techniques to advance their respective domains.

**Decision-focused learning** (DFL; Mandi et al., 2024). DFL, also termed as "predict-then-optimize", aims to predict unknown parameters for an optimization problem using ML model in an end-to-end paradigm. A recent popular work of this field is Mandi et al. (2022), which utilizes LTR losses that preserve the correct order of solutions in the discrete feasible space to train a better parameter-predicting model.

**Preference-based reinforcement learning** (PbRL; Christiano et al., 2017). The goal of PbRL is to infer reward functions from human feedback in the form of preferences or rankings over demonstrated behaviors. Memarian et al. (2021) define a preference oracle to measure the total order equivalency and use pairwise ranking loss to train a reward model for the sparse-reward environments.

**Language model alignment** (Shen et al., 2023). The objective of language model alignment is to let the models align with human preferences. Song et al. (2024) adopt LTR techniques to process human preference rankings of varying lengths, while Liu et al. (2024) formulate the problem as a listwise ranking problem, which can learn more efficiently from a given ranked list of response.

However, our work differs from these works in both motivation and methodology. We focus on offline MBO and investigate the root cause of the OOD issue, which is widely-studied in this field but still remains. We provide a systematic analysis of the OOD issue, propose the AUPCC metric for quantification, develop a ranking-based framework, and verify its effectiveness through theoretical analysis and comprehensive experiments.

## B PREVALENT SETTINGS OF $\phi$, $N(\phi)$, AND $C_{\mathcal{A}}(\phi)$ IN THEOREM 2

In this section, we introduce some settings of $\phi$, $N(\phi)$, and $C_{\mathcal{A}}(\phi)$ in Theorem 2, as shown in Lan et al. (2009).

In Theorem 2, $\phi$ is an increasing and strictly positive transformation function, which maps the output of the surrogate model or the score to a positive real number. Recall that $B$ represents the upper bound of the weight norm $\|\mathbf{w}\|$ of the linear function class $\mathcal{F} = \{\mathbf{x} \to \mathbf{w}^\top \mathbf{x} \mid \|\mathbf{w}\| \leq B\}$ where the ranking model $f$ to be learned is from, and $M$ is the upper bound of the norm of designs $\|\mathbf{x}\|$ in design space $\mathcal{X}$. It is usually represented as a:

- Linear function: $\phi_L(z) = az + b$, $z \in [-BM, BM]$, where $a > 0$ and $b > aBM$;
- Exponential function: $\phi_E(z) = \exp(az)$, $z \in [-BM, BM]$, where $a > 0$;
- Sigmoid function: $\phi_S(z) = \frac{1}{1+\exp(-az)}$, $z \in [-BM, BM]$, where $a > 0$.

Following Lan et al. (2009), we introduce some settings based on the above definition of $\phi$ in Table 4. For detailed derivation for $C_{\mathcal{A}}(\phi)$, please refer to Lan et al. (2009).

## C PROBABLE APPROACHES AND DIFFICULTIES FOR THEORETICAL ANALYSIS

In this section, we first further discuss the probable approaches and difficulties for direct theoretical analysis for ranking-based framework for offline MBO. Although it is challenging, we still find a

Table 4: $N(\phi)$ and $C_{\mathcal{A}}(\phi)$ for LTR algorithms $\mathcal{A}$ (e.g., RankCosine (Qin et al., 2008) or ListNet (Cao et al., 2007)) on different definitions of $\phi$.

| $\phi$ | $N(\phi)$ | $C_{RankCosine}(\phi)$ | $C_{ListNet}(\phi)$ |
|---|---|---|---|
| $\phi_L(z) = az + b$ | $a$ | $\frac{\sqrt{m}}{2(b-aBM)}$ | $\frac{2m!}{(b-aBM)(\log m + \log\frac{b+aBM}{b-aBM})}$ |
| $\phi_E(z) = \exp(az)$ | $a\exp(aBM)$ | $\frac{\sqrt{m}\exp(aBM)}{2}$ | $\frac{2m!\exp(aBM)}{\log m + 2aBM}$ |
| $\phi_S(z) = \frac{1}{1+\exp(-az)}$ | $\frac{a(1+\exp(aBM))}{(1+\exp(-aBM))^2}$ | $\frac{\sqrt{m}(1+\exp(aBM))}{2}$ | $\frac{2m!(1+\exp(aBM))}{\log m + aBM}$ |

counterexample that shows the robustness of LTR losses over MSE. Then, to enhance understanding of the counterexample, we conduct a quantitative experiment to demonstrate this.

## C.1 PROBABLE APPROACHES AND DIFFICULTIES

In this subsection, firstly, we revisit our motivation to leverage LTR techniques for offline MBO. Then, we propose some probable approaches and difficulties for theoretical analysis.

Note that learning to rank samples correctly is a weaker condition than learning to minimize MSE, since MSE commands for both order preserving and value matching. Besides, the equivalence in Theorem 1 shows that the weaker condition, order preserving, is sufficient for offline MBO, which motivates the proposal of directly learn the ranking information by leveraging LTR techniques.

Thus, a intuitive question according to generalization analysis for offline MBO is: *In which scenarios does the model learned with LTR generalize better on some ranking measures than that learned with MSE on OOD regions?* Unfortunately, such theoretical support or evidence cannot be found even in the field of LTR, which is also illustrated in Section 1 of Chapelle et al. (2010). Below we briefly present the most promising approach we explored and the difficulties we face.

- Try to find a special function class $\mathcal{F}$, from which the ranking model $\hat{f}$ to be learned is, such that models learned with LTR techniques have an upper bound guarantee on some ranking measure while models trained with MSE do not. Formally, let $R$ be a ranking measure (which can be the expected risk of a specific ranking loss or a ranking metric, e.g., NDCG), and denote the empirical risk of model trained with LTR and that trained with MSE as $\hat{R}_{LTR}$ and $\hat{R}_{MSE}$, respectively. For ease of exposition, $R$ here refers to the expected risk in Theorem 2. From Theorem 2, the upper bound of $R$ and $\hat{R}_{LTR}$ has a convergence rate of $\mathcal{O}(\frac{1}{\sqrt{n}})$. Then, if we could find a function class $\mathcal{F}$ such that $R - \hat{R}_{MSE}$ always has a slower convergence rate, i.e., $R - \hat{R}_{MSE} \geq \mathcal{O}(\frac{1}{\sqrt{n}})$, we can show that models learned with MSE are worse than that learned with LTR. However, such an analysis can be difficult because: 1) There is no theoretical evidence to show the generalization bound on ranking by optimizing MSE. 2) Most generalization bound analysis in LTR assume i.i.d (as Theorem 2 in our paper), while OOD analysis in LTR is quite limited.

- Identify a special case that supports this intuition. Assume that the function class $\mathcal{F}$ is a linear function class and the offline data is drawn from a ground-truth function $f$ with long-tailed noise on the objective value. Models trained with MSE are susceptible to heavy-tailed noise, as the mean of $y$ is heavily influenced in regions with such noise. In contrast, models trained with pairwise ranking loss demonstrate greater stability in such scenarios. An illustrative example could be as follows. Assume that the ground-truth function is $f(x) = x^2$ and the offline dataset $\mathcal{D} = \{(1, 1), (1.9, 3.7), (2.1, 4.5), (2, -12)\}$ where $(2, -12)$ suffers from the heavy-tailed noise. Models trained with MSE and a representative of the pairwise ranking loss, RankCosine (Qin et al., 2008), are shown in Figure 3. From Figure 3, the model trained with MSE would exhibit negative correlation, while that trained with LTR would demonstrate positive correlation, which shows that the model trained with LTR is more robust. However, such counterexamples are still based on strong assumptions. A well-constructed example with theoretical support remains unexplored.

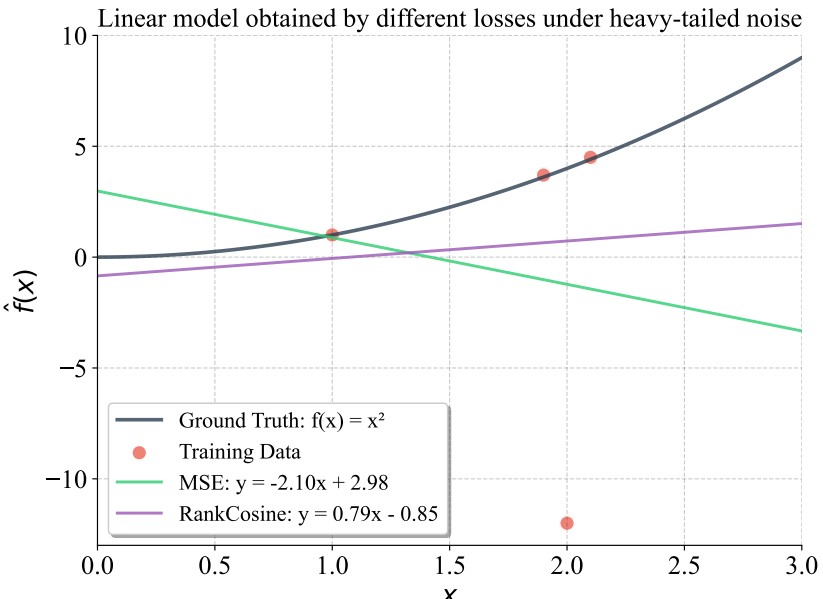

Figure 3: Plot of the ground-truth function $f(x) = x^2$, the training data suffered from heavy-tailed noise, the linear model learned with MSE (green), and the linear model learned with RankCosine. Here the model trained with MSE exhibits negative correlation, while that trained with LTR demonstrates positive correlation, which shows that the model trained with LTR is more robust.

## C.2 ADDITIONAL EXPERIMENTS IN HEAVY-TAILED NOISY SCENARIOS

In this subsection, we conduct additional quantitative experiments to support the counterexample mentioned in the above subsection.

Following the assumption in Appendix C.1, the ranking model $\hat{f}$ to be learned is from the linear function class. Specifically, given a dataset $\mathcal{D} = \{(x_i, y_i)\}_{i=1}^N$, we want to train a linear model $\hat{f}(x) = wx + b$ based on two loss functions, e.g., MSE and RankCosine (Qin et al., 2008), the representative of pairwise ranking losses. Details of how to obtain the linear model trained with these two losses are as follows.

- MSE. The linear model trained with MSE has a closed-formed solution using the Least Squares Method. Formally, let an augmented matrix $X = [\mathbf{1}, [x_1, x_2, \cdots, x_N]^\top]$, $\mathbf{y} = [y_1, y_2, \cdots, y_N]^\top$, and $\theta = [w, b]^\top$, and we can obtain that $\theta = (X^\top X)^{-1} X^\top \mathbf{y}$ (see Chapter 3.1.1 in Bishop (2006)).

- RankCosine: There is no closed-formed solution due to non-linear operations (specifically, vector normalization operator when calculating RankCosine). Hence, we use Adam optimizer (Kingma & Ba, 2015) with a learning rate $1 \times 10^{-3}$ to search 1000 epochs for the optimal value for $w$ and $b$.

We set the ground-truth function to be a quadratic function $f(x) = x^2$ for ease of demonstration, which is increasing and requires $\hat{f}$ having a positive $w$. We assume that the training data is drawn from $[0, 3]$ for better visualization. As for the noise, we initiate the heavy-tailed noises from a Student's t-distribution $g(t) = \frac{\Gamma(\frac{\nu+1}{2})}{\sqrt{\nu\pi}\,\Gamma(\frac{\nu}{2})}\left(1 + \frac{t^2}{\nu}\right)^{-\frac{\nu+1}{2}}$ with the degrees of freedom $\nu = 2$, and change their magnitude controlled by a scale $\alpha = 15$. Besides, to influence the increasing trend, we assume that the heavy-tailed noise is positive for points with $x \in [0, 1.5]$ and negative for points with $x \in (1.5, 3]$. Each training point has a probability of $p = 0.2$ to suffer from the noise.

We first present the detailed results of the illustrative example mentioned in Appendix C.1. In Figure 3, we visualize the ground-truth function, training data, and the linear models trained with MSE and RankCosine. We can observe that the model learned from MSE exhibits a negative correlation, but the model learned from RankCosine can demonstrate a positive correlation.

To further verify the robustness of ranking losses, we increase the dataset size to 100, and vary the scale of noise $\alpha \in \{10, 15, 20, 50, 100\}$ while the probability of adding noise is fixed at $p = 0.2$. We report the calculated values of $w$ learned with MSE (denoted as $w_{MSE}$) and that learned with RankCosine (denoted as $w_{RankCosine}$) according to different $\alpha$s in Table C.2. From Table C.2, all

Table 5: Values of weight $w$ obtained by learning MSE (denoted as $w_{MSE}$) and those obtained by learning RankCosine (denoted as $w_{RankCosine}$) with varying noise scale $\alpha$. Here, **Violet** denote positive weights, which satisfies the requirements of the ground-truth function $f(x) = x^2$ for a increasing linear ranking model.

| Noise scale $\alpha$ | $w_{MSE}$ | $w_{RankCosine}$ |
|:---:|:---:|:---:|
| 10 | -1.68 | **0.88** |
| 15 | -0.75 | **0.90** |
| 20 | -2.98 | **0.74** |
| 50 | -14.99 | **0.94** |
| 100 | -10.05 | **0.98** |

values of $w_{RankCosine}$ are positive while those of $w_{MSE}$ are all negative and become substantially worse when the scale of noise $\alpha$ goes larger, which demonstrates the stronger stability of the LTR loss against heavy-tailed noise with different strengths.

We also vary the probability of adding noise $p \in \{0.1, 0.2, \cdots, 1.0\}$ while the scale of noise is fixed at $\alpha = 15$. The corresponding values of $w$ are shown in Table 6.

Table 6: Values of weight $w$ obtained by learning MSE (denoted as $w_{MSE}$) and those obtained by learning RankCosine (denoted as $w_{RankCosine}$) with varying noise probability $p$. Here, **Violet** denote positive weights, which satisfies the requirements of the ground-truth function $f(x) = x^2$ for a increasing linear ranking model.

| Noise probability $p$ | $w_{MSE}$ | $w_{RankCosine}$ |
|:---:|:---:|:---:|
| 0.1 | **1.61** | **0.86** |
| 0.2 | -0.75 | **0.90** |
| 0.3 | -4.53 | **1.01** |
| 0.4 | -8.46 | **0.88** |
| 0.5 | -7.76 | **1.02** |
| 0.6 | -10.73 | **0.95** |
| 0.7 | -12.87 | **0.84** |
| 0.8 | -17.28 | **0.98** |
| 0.9 | -20.21 | **0.98** |
| 1.0 | -22.66 | **0.95** |

From the results in Table 6, only when the noise probability $p = 0.1$, $w_{MSE}$ is positive, while in other situations it is negative and it becomes quite bad as $p$ increases. In contrast, $w_{RankCosine}$ remains a positive value near 1 as the noise probability $p$ increases from 0.1 to 1, showing impressive robustness against such heavy-tailed noise with wide coverage.

Results from both Table C.2 and Table 6 strongly demonstrate the robustness of pairwise ranking loss (i.e., RankCosine) over MSE on the ranking performance in a scenario where $y$ suffers from a heavy-tailed noise, which delivers a better understanding on the advantage of LTR losses in OOD ranking performance. Combining with the stated equivalence of an order-preserving surrogate model shown in Theorem 1, the ranking loss is suitable for offline MBO due to its more robust ranking performance.

## D    DETAILS OF DIFFERENT RANKING LOSSES

In this section, we introduce details of the different ranking losses in this paper, including traditional and recently prevalent losses. We study different types of ranking losses in this paper, including pointwise (Crammer & Singer, 2001), pairwise (Köppel et al., 2019), and listwise losses (Xia

et al., 2008). Formally, given a list $\mathbf{X}$ of designs and the list $\mathbf{y}$ of their corresponding scores, let $\hat{f}_{\boldsymbol{\theta}}(\mathbf{X}) = [\hat{f}_{\boldsymbol{\theta}}(\mathbf{x}_1), \hat{f}_{\boldsymbol{\theta}}(\mathbf{x}_2), \ldots, \hat{f}_{\boldsymbol{\theta}}(\mathbf{x}_m)]^{\top}$ be the predicted scores.

For the pointwise losses, we consider:

- SigmoidCrossEntropy (SCE): a widely used pointwise loss: $l(\mathbf{y}, \hat{f}_{\boldsymbol{\theta}}(\mathbf{X})) = \sum_{i=1}^{m} \left( -y_i \hat{f}_{\boldsymbol{\theta}}(\mathbf{x}_i) + \log(1 + \exp(\hat{f}_{\boldsymbol{\theta}}(\mathbf{x}_i))) \right)$.

- BinaryCrossEntropy (BCE): a common pointwise loss considering in a binary classification manner, and we consider its variant with a logits input: $l(\mathbf{y}, \hat{f}_{\boldsymbol{\theta}}(\mathbf{X})) = -\sum_{i=1}^{m} \left[ y_i \cdot \log(\sigma(\hat{f}_{\boldsymbol{\theta}}(\mathbf{x_i}))) + (1 - y_i) \cdot \log(1 - \sigma(\hat{f}_{\boldsymbol{\theta}}(\mathbf{x_i}))) \right]$, where $\sigma(\cdot)$ is the sigmoid function.

- Mean Square Error (MSE): a popular pointwise loss aiming to fit the target values: $l(\mathbf{y}, \hat{f}_{\boldsymbol{\theta}}(\mathbf{X})) = \sum_{i=1}^{m} (y_i - \hat{f}_{\boldsymbol{\theta}}(\mathbf{x}_i))^2$. Note that the difference of RaM combined with MSE from the regression-based model mainly reflects in the different modeling of the training data.

For the pairwise losses, we consider:

- RankNet (Burges et al., 2005): a popular pairwise loss: $l(\mathbf{y}, \hat{f}_{\boldsymbol{\theta}}(\mathbf{X})) = \sum_{y_i > y_j} \log \left( 1 + \exp(\hat{f}_{\boldsymbol{\theta}}(\mathbf{x}_i) - \hat{f}_{\boldsymbol{\theta}}(\mathbf{x}_j)) \right)$.

- LambdaRank (Burges et al., 2006; Wang et al., 2018): a pairwise loss with $\Delta$NDCG weight: $l(\mathbf{y}, \hat{f}_{\boldsymbol{\theta}}(\mathbf{X})) = \sum_{y_i > y_j} \Delta NDCG(i, j) \log_2 \left( 1 + \exp(-\alpha(\hat{f}_{\boldsymbol{\theta}}(\mathbf{x}_i) - \hat{f}_{\boldsymbol{\theta}}(\mathbf{x}_j))) \right)$, where $\alpha$ is a smooth parameter and $\Delta$NDCG is the absolute difference between the values of the Normalized Discounted Cumulative Gain (NDCG), a widely used metric in LTR (Järvelin & Kekäläinen, 2000; 2002), when the surrogate model swap the predictions of the two designs, $\mathbf{x}_i$ and $\mathbf{x}_j$, and thus swap their positions in the ranked list.

- RankCosine (Qin et al., 2008): a classical pairwise loss based on cosine similarity: $l(\mathbf{y}, \hat{f}_{\boldsymbol{\theta}}(\mathbf{X})) = 1 - \mathbf{y} \cdot \hat{f}_{\boldsymbol{\theta}}(\mathbf{X}) / (\|\mathbf{y}\| \cdot \|\hat{f}_{\boldsymbol{\theta}}(\mathbf{X})\|)$.

For the pairwise losses, we consider:

- Softmax (Cao et al., 2007; Bruch et al., 2019a): a popular listwise loss: $l(\mathbf{y}, \hat{f}_{\boldsymbol{\theta}}(\mathbf{X})) = -\sum_{i=1}^{m} y_i \log \frac{\exp(\hat{f}_{\boldsymbol{\theta}}(\mathbf{x}_i))}{\sum_{j=1}^{m} \exp(\hat{f}_{\boldsymbol{\theta}}(\mathbf{x}_i))}$.

- ListNet (Cao et al., 2007): a classical listwise loss minimizing the cross-entropy between the predicted ranking distribution and the true ranking distribution: $l(\mathbf{y}, \hat{f}_{\boldsymbol{\theta}}(\mathbf{X})) = -\sum_{j=1}^{m} \frac{\exp(y_j)}{\sum_{i=1}^{m} \exp(y_i)} \log \frac{\exp(\hat{f}_{\boldsymbol{\theta}}(\mathbf{x}_j))}{\sum_{i=1}^{m} \exp(\hat{f}_{\boldsymbol{\theta}}(\mathbf{x}_i))}$.

- ListMLE (Xia et al., 2008): a widely used listwise loss based on the Plackett-Luce model (Marden, 1995): $l(\mathbf{y}, \hat{f}_{\boldsymbol{\theta}}(\mathbf{X})) = -\sum_{i=1}^{m} \log \frac{\exp(\hat{f}_{\boldsymbol{\theta}}(\mathbf{x}_{\pi(i)}))}{\sum_{j=i}^{m} \exp(\hat{f}_{\boldsymbol{\theta}}(\mathbf{x}_{\pi(j)}))}$, where $\pi$ is the permutation derived from the true ranking labels $y$, $\mathbf{x}_{\pi(i)}$ represents the item at the $i$-th position in the true ranking.

- ApproxNDCG (Qin et al., 2010a; Bruch et al., 2019b): a listwise that is a differentiable approximation of NDCG: $l(\mathbf{y}, \hat{f}_{\boldsymbol{\theta}}(\mathbf{X})) = -\frac{1}{DCG(\pi^*, \mathbf{y})} \sum_{i,r=1}^{m} \frac{2^{y_i} - 1}{\log_2(1 + \pi_{\hat{f}_{\boldsymbol{\theta}}}(i))}$, where $\pi^*$ is the optimal permutation that ranks items by $\mathbf{y}$, $DCG(\pi^*, \mathbf{y})$ represents the Discounted Cumulative Gain (DCG; Järvelin & Kekäläinen, 2000; 2002) of the ideal ranking given $\mathbf{y}$, and $\pi_{\hat{f}_{\boldsymbol{\theta}}}(i) = \frac{1}{2} + \sum_j \text{Sigmoid}(\frac{\hat{f}_{\boldsymbol{\theta}}(\mathbf{x}_i) - \hat{f}_{\boldsymbol{\theta}}(\mathbf{x}_j)}{T})$ with $T$ a smooth parameter.

We excluded NeuralNDCG (Pobrotyn & Bialobrzeski, 2021), a recently proposed listwise loss using neural sort techniques to approximate NDCG, due to its high memory requirements.

# E    DETAILED EXPERIMENTAL SETTINGS

## E.1    DETAILED EXPERIMENTAL SETTINGS OF FIGURE 2

In this experiment, we select five surrogate models: a gradient-ascent baseline and four state-of-the-art approaches, COMs (Trabucco et al., 2021), IOM (Qi et al., 2022), ICT (Yuan et al., 2023), and Tri-Mentoring (Chen et al., 2023a). These models are chosen due to their common characteristic of employing standard gradient-ascent to obtain the final design. While BDI (Chen et al., 2022) and Match-OPT (Hoang et al., 2024) also utilize gradient-ascent for design generation, we exclude BDI for its intractable model, which is built with JAX (Bradbury et al., 2018), and Match-OPT for its time-intensive training procedure.

We follow the default setting as in Chen et al. (2023a); Yuan et al. (2023) to prepare training data and set the hyper-parameters in Equation 1 to search inside the model. For discrete tasks, in order to map the design space to a continuous one, we transform the discrete designs into real-valued logits of a categorical distribution, which is provided in Trabucco et al. (2021; 2022). We use z-score method to normalize both the designs and scores for a better training. After the model is trained, we use Adam optimizer (Kingma & Ba, 2015) to conduct gradient ascent. For discrete tasks, we set $\eta = 1 \times 10^{-1}$ and $T = 100$, and for continuous tasks, we set $\eta = 1 \times 10^{-3}$ and $T = 200$.

Following Chen et al. (2023a), we construct an OOD dataset by selecting the high-scoring designs that are excluded for the training data in Design-Bench (Trabucco et al., 2022). In Design-Bench, the training dataset is selected as the bottom performing $x\%$ in the entire collected dataset, (i.e., $x = 40, 50, 60$). Note that the open-source repository[4] provides an API to access the entire dataset. We identify the excluded $(100 - x)\%$ high-scoring data to comprise the OOD dataset for analysis, except for TF-Bind-10 (Barrera et al., 2016) task, whose excluded $(100 - x)\%$ high-scoring data contains 4161482 samples and is too large for AUPRC evaluation. Thus, we randomly sample 30000 samples from the $(100 - x)\%$ data to construct the OOD dataset for TF-Bind-10 task.

## E.2    EXCLUDED DESIGN-BENCH TASKS

Following prior works (Krishnamoorthy et al., 2023; Mashkaria et al., 2023; Yun et al., 2024; Yu et al., 2024), we exclude three tasks in Design-Bench (Trabucco et al., 2022) for evaluation, including Hopper (Brockman et al., 2016), ChEMBL (Gaulton et al., 2012), and synthetic NAS tasks on CIFAR10 (Hinton et al., 2012). As noted in prior works, this is a bug for the implementation of Hopper in Design-Bench (see `https://github.com/brandontrabucco/design-bench/issues/8#issuecomment-1086758113` for details). For the ChEMBL task, we exclude it because almost all methods produce the same results, as shown in Mashkaria et al. (2023); Krishnamoorthy et al. (2023), which is not suitable for comparison. We also exclude NAS due to its high computation cost for exact evaluation over multiple seeds, which is beyond our budget.

## E.3    EXCLUDED OFFLINE MBO ALGORITHMS

We exclude NEMO (Fu & Levine, 2021) since there is no open-source implementation. We also exclude concurrent works, DEMO (Yuan et al., 2024) and LEO (Yu et al., 2024), since they are not yet peer-reviewed and lack an open-source implementation at the time of our initial submission. For BOSS (Dao et al., 2024b), we exclude it since it is a general trick that can be applied to any regression-based forward method, instead of a single proposed methods.

## E.4    DETAILED EXPERIMENTAL SETTINGS OF MAIN RESULTS IN TABLE 2

We set the size $n$ of training dataset to 10000, and the list length $m = 1000$. To make a fair comparison to regression-based methods, following Trabucco et al. (2021; 2022); Chen et al. (2023a); Yuan et al. (2023), we model the surrogate model $\hat{f}_{\boldsymbol{\theta}}$ as a simple multilayer perceptron (MLP) with two hidden layers of size 2048 using PyTorch (Paszke et al., 2019). We use ReLU as activation functions. RankCosine (Qin et al., 2008) and ListNet (Cao et al., 2007) is used as two main loss

---

[4]`https://github.com/brandontrabucco/design-bench`

functions in our experiments. Our implementation of different loss functions is either inherited from Pobrotyn et al. (2020)[5] or implemented by ourselves.

We split the dataset into a training set and a validation set of the ratio $8 : 2$. The model is trained for $N_0 = 200$ epochs and is optimized using Adam (Kingma & Ba, 2015) with a learning rate of $3 \times 10^{-4}$ and a weight decay coefficient of $1 \times 10^{-5}$, and the model with minimal validation loss among $N_0$ epochs serves as the final model.

After the model is trained, we fix the model parameters and normalize the output values, then following Chen et al. (2023a); Yuan et al. (2023), we set $\eta = 1 \times 10^{-3}$ and $T = 200$ for continuous tasks, and $\eta = 1 \times 10^{-1}$ and $T = 100$ for discrete tasks to search for the final design.

For baselines methods and CbAS (Brookes et al., 2019), MINs (Kumar & Levine, 2020), COMs (Trabucco et al., 2021) in Table 2, we use the open-source baselines implementations from the source code of Design-Bench[6]. For other offline MBO methods (DDOM (Krishnamoorthy et al., 2023)[7], BONET (Mashkaria et al., 2023)[8], GTG (Yun et al., 2024)[9], RoMA (Yu et al., 2021)[10], IOM (Qi et al., 2022)[11], BDI (Chen et al., 2022)[12], ICT (Yuan et al., 2023)[13], Tri-Mentoring (Chen et al., 2023a)[14], PGS (Chemingui et al., 2024)[15], FGM (Kuba et al., 2024b)[16], Match-OPT (Hoang et al., 2024)[17]), we use the open-source implementation provided in their papers and use their hyper-parameter settings, except for DDOM and BONET, where we modify the evaluation budget $k$ from 256 to 128 following the protocol of other works. A brief review of offline MBO methods is also provided in Appendix A.1.

### E.5 Detailed Experimental Settings of Table 3

In this experiment, for a fair comparison of MSE and ListNet, we do not adopt the data augmentation method, instead, we use the naïve approach introduced in Section 3.3, viewing a batch of designs as a list to be ranked.

We choose baselines methods that optimize a trained model, BO-$q$EI (Garnett, 2023), CMA-ES (Hansen, 2016), REINFORCE (Williams, 1992), and Gradient Ascent, two backward approach provided in Trabucco et al. (2022), CbAS (Brookes et al., 2019) and MINs (Kumar & Levine, 2020), and three state-of-the-art forward methods that can replace MSE with ListNet, Tri-Mentoring (Chen et al., 2023a), PGS (Chemingui et al., 2024), and Match-OPT (Hoang et al., 2024). Note that the model trained with ranking loss has different prediction scales as regression-based models, as discussed in 3.3. We exclude many forward methods due to the inapplicability of directly replacing MSE with ListNet. For example, COMs (Trabucco et al., 2021), RoMA (Yu et al., 2021), IOM (Qi et al., 2022) use the prediction values to calculate the loss function, where the changing scales of predictions could influence the scales of the loss values, while BDI (Chen et al., 2022) and ICT (Yuan et al., 2023) assign weight to each sample, thus MSE in these methods cannot be directly replaced with a ranking loss like ListNet.

In order to adapt the same parameters of the online optimizers (e.g., BO-$q$EI, Gradient Ascent) that optimize the trained model for a fair comparison, we also perform an output adaptation for ranking-based model after it is trained.

All the replacements are conducted fixing their open-source codes by replacing MSE with ListNet when training the forward model.

---

[5] https://github.com/allegro/allRank

[6] https://github.com/brandontrabucco/design-baselines

[7] https://github.com/siddarthk97/ddom

[8] https://github.com/siddarthk97/bonet

[9] https://github.com/dbsxodud-11/GTG

[10] https://github.com/sihyun-yu/RoMA

[11] https://anonymous.4open.science/r/IOMsubmit-265E

[12] https://github.com/GGchen1997/BDI

[13] https://github.com/mila-iqia/Importance-aware-Co-teaching

[14] https://github.com/GGchen1997/parallel_mentoring

[15] https://github.com/yassineCh/PGS

[16] https://colab.research.google.com/drive/1qt4M3C35bvjRHPIpBxE3zPc5zvX6AAU4?usp=sharing

[17] https://github.com/azzafadhel/MatchOpt

Table 7: 50th percentile normalized score in Design-Bench, where the best and runner-up results on each task are **Blue** and **Violet**. $\mathcal{D}$(best) denotes the best score in the offline dataset.

| Method | Ant | D'Kitty | Superconductor | TF-Bind-8 | TF-Bind-10 | Mean Rank |
|---|---|---|---|---|---|---|
| $\mathcal{D}$(best) | 0.565 | 0.884 | 0.400 | 0.439 | 0.467 | / |
| BO-$q$EI | 0.568 ± 0.000 | 0.883 ± 0.000 | 0.311 ± 0.019 | 0.439 ± 0.000 | 0.467 ± 0.000 | 12.6 / 22 |
| CMA-ES | -0.041 ± 0.004 | 0.684 ± 0.017 | 0.377 ± 0.009 | 0.539 ± 0.017 | 0.482 ± 0.009 | 12.6 / 22 |
| REINFORCE | 0.124 ± 0.042 | 0.460 ± 0.209 | 0.457 ± 0.020 | 0.466 ± 0.023 | 0.464 ± 0.009 | 15.7 / 22 |
| Grad. Ascent | 0.136 ± 0.016 | 0.581 ± 0.128 | 0.471 ± 0.017 | 0.582 ± 0.027 | 0.470 ± 0.004 | 11.2 / 22 |
| Grad. Ascent Mean | 0.185 ± 0.012 | 0.718 ± 0.037 | **0.481 ± 0.023** | **0.630 ± 0.033** | 0.470 ± 0.005 | 9.2 / 22 |
| Grad. Ascent Min | 0.187 ± 0.012 | 0.714 ± 0.040 | **0.480 ± 0.022** | **0.628 ± 0.025** | 0.470 ± 0.004 | 9.6 / 22 |
| CbAS | 0.385 ± 0.027 | 0.740 ± 0.023 | 0.121 ± 0.014 | 0.422 ± 0.022 | 0.457 ± 0.006 | 18.4 / 22 |
| MINs | **0.640 ± 0.029** | 0.886 ± 0.006 | 0.332 ± 0.014 | 0.407 ± 0.014 | 0.465 ± 0.006 | 12.0 / 22 |
| DDOM | 0.598 ± 0.030 | 0.829 ± 0.050 | 0.313 ± 0.017 | 0.416 ± 0.023 | 0.464 ± 0.006 | 14.4 / 22 |
| BONET | **0.795 ± 0.039** | **0.906 ± 0.008** | 0.334 ± 0.032 | 0.476 ± 0.149 | 0.452 ± 0.050 | 10.6 / 22 |
| GTG | 0.593 ± 0.022 | **0.889 ± 0.002** | 0.350 ± 0.023 | 0.542 ± 0.038 | 0.458 ± 0.008 | 9.4 / 22 |
| COMs | 0.532 ± 0.020 | 0.882 ± 0.002 | 0.376 ± 0.067 | 0.513 ± 0.019 | 0.474 ± 0.014 | 9.3 / 22 |
| RoMA | 0.193 ± 0.017 | 0.344 ± 0.097 | 0.368 ± 0.012 | 0.520 ± 0.074 | **0.516 ± 0.004** | 12.0 / 22 |
| IOM | 0.459 ± 0.024 | 0.829 ± 0.022 | 0.291 ± 0.059 | 0.490 ± 0.055 | 0.467 ± 0.000 | 14.9 / 22 |
| BDI | 0.569 ± 0.000 | 0.876 ± 0.000 | 0.389 ± 0.022 | 0.595 ± 0.000 | 0.429 ± 0.000 | 9.4 / 22 |
| ICT | 0.550 ± 0.028 | 0.875 ± 0.006 | 0.333 ± 0.018 | 0.547 ± 0.041 | **0.499 ± 0.012** | 9.2 / 22 |
| Tri-Mentoring | 0.548 ± 0.013 | 0.870 ± 0.002 | 0.363 ± 0.019 | 0.619 ± 0.009 | 0.491 ± 0.001 | **8.0 / 22** |
| PGS | 0.190 ± 0.030 | 0.885 ± 0.001 | 0.233 ± 0.033 | 0.503 ± 0.041 | 0.386 ± 0.177 | 16.0 / 22 |
| FGM | 0.532 ± 0.039 | 0.871 ± 0.017 | 0.353 ± 0.058 | 0.540 ± 0.117 | 0.466 ± 0.004 | 12.1 / 22 |
| Match-OPT | 0.587 ± 0.008 | 0.887 ± 0.001 | 0.381 ± 0.038 | 0.435 ± 0.017 | 0.471 ± 0.013 | **8.0 / 22** |
| **RaM-RankCosine (Ours)** | 0.566 ± 0.012 | 0.881 ± 0.003 | 0.356 ± 0.013 | 0.544 ± 0.043 | 0.462 ± 0.006 | 11.0 / 22 |
| **RaM-ListNet (Ours)** | 0.579 ± 0.014 | 0.888 ± 0.003 | 0.359 ± 0.013 | 0.552 ± 0.032 | 0.467 ± 0.009 | **7.4 / 22** |

# F  ADDITIONAL EXPERIMENTS

In this section, we provide additional experimental results mentioned in Section 4.

## F.1  50TH PERCENTILE RESULTS ON DESIGN-BENCH

Following the evaluation protocol in Trabucco et al. (2022), to validate the robustness of our proposed method, we also provide the detailed results of 50th percentile results in Table 7.

In Table 7, we can observe although RaM combined with RankCosine performs not so well on $50^{th}$ percentile results, RaM combined with ListNet, which is the best methods in our main experimental results (Table 2), also obtains a best average rank of 7.4 among 22 methods.

## F.2  RESULTS OF DIFFERENT RANKING LOSSES

We compare a wide range of ranking losses that combined with RaM in the context of offline MBO, including three types of pointwise, pairwise, and listwise losses. Details of these ranking losses are provided in Appendix D, and experimental results of 100th percentile normalized score in Design-Bench are provided in Table 8.

Table 8: 100th percentile normalized score of RaM combined with different ranking losses in Design-Bench. The best and runner-up results on each task are **Blue** and **Violet**. $\mathcal{D}$(best) denotes the best score in the offline dataset.

| Type | Method | Ant | D'Kitty | Superconductor | TF-Bind-8 | TF-Bind-10 | Mean Rank |
|---|---|---|---|---|---|---|---|
| / | $\mathcal{D}$(best) | 0.565 | 0.884 | 0.400 | 0.439 | 0.467 | / |
| Pointwise | RaM-SCE | 0.928 ± 0.012 | 0.953 ± 0.012 | 0.502 ± 0.013 | 0.820 ± 0.065 | 0.662 ± 0.026 | 6.9 / 10 |
| | RaM-BCE | 0.925 ± 0.014 | 0.950 ± 0.009 | 0.501 ± 0.012 | 0.825 ± 0.065 | 0.656 ± 0.021 | 8.3 / 10 |
| | RaM-MSE | 0.933 ± 0.032 | **0.957 ± 0.013** | 0.507 ± 0.028 | 0.962 ± 0.031 | 0.674 ± 0.044 | 3.9 / 10 |
| Pairwise | RaM-RankNet | 0.921 ± 0.033 | 0.955 ± 0.008 | 0.510 ± 0.032 | 0.962 ± 0.030 | **0.676 ± 0.037** | 4.4 / 10 |
| | RaM-LambdaRank | 0.918 ± 0.020 | 0.949 ± 0.010 | **0.528 ± 0.020** | 0.962 ± 0.020 | 0.650 ± 0.039 | 6.8 / 10 |
| | RaM-RankCosine | **0.940 ± 0.028** | 0.951 ± 0.017 | 0.514 ± 0.026 | **0.982 ± 0.012** | **0.675 ± 0.049** | **3.2 / 10** |
| Listwise | RaM-Softmax | 0.932 ± 0.014 | 0.954 ± 0.011 | 0.509 ± 0.028 | 0.918 ± 0.039 | 0.489 ± 0.115 | 6.2 / 10 |
| | RaM-ListNet | **0.949 ± 0.025** | **0.962 ± 0.015** | **0.517 ± 0.029** | **0.981 ± 0.012** | 0.670 ± 0.035 | **2.0 / 10** |
| | RaM-ListMLE | 0.930 ± 0.032 | 0.953 ± 0.012 | 0.484 ± 0.022 | 0.966 ± 0.020 | 0.656 ± 0.041 | 6.0 / 10 |
| | RaM-ApproxNDCG | 0.926 ± 0.031 | 0.952 ± 0.004 | 0.507 ± 0.010 | 0.936 ± 0.069 | 0.551 ± 0.058 | 7.3 / 10 |

We find that MSE performs the best in all of 3 pointwise losses, RankCosine (Qin et al., 2008) outperforms other pairwise losses, and ListNet (Cao et al., 2007) obtains the highest average rank among listwise losses. Note that prevalent ranking losses such as ApproxNDCG (Bruch et al., 2019b) do not perform well in RaM. This might due to the simplicity of MLP, which cannot absorb complex information of conveyed by the trending powerful loss functions (Qin et al., 2021; Pobrotyn et al., 2020). However in this work, we parameterize the surrogate model as a simple MLP for a fair comparison to the regression-based methods, and we will consider more complex modeling in our future work.

## F.3  ABLATION STUDIES RESULTS ON MAIN MODULES

To better validate the effectiveness of the two moduels, *data augmentation* and *output adaptation*, of our method, we perform ablation studies based on the top-performing loss functions shown in Table 8: MSE for pointwise loss, RankCosine for pairwise loss, and ListNet for listwise loss. The results in Table 9 show that for each considered loss, RaM with data augmentation performs better than the naïve approach which treats a batch of the dataset as a list to rank. The results in Table 10 show the benefit of using output adaptation. All of these ablation studies provide strongly positive support to the effectiveness of these two modules.

Table 9: Ablation studies on data augmentation, considering learning with MSE, RankCosine, and ListNet, which are the best-performing pointwise, pairwise, and listwise loss, respectively, as shown in Table 8. For each combination of loss and task, the better performance is **Bolded**. $\mathcal{D}$(best) denotes the best score in the offline dataset.

| Method | Ant | D'Kitty | Superconductor | TF-Bind-8 | TF-Bind-10 | Mean Rank |
|---|---|---|---|---|---|---|
| $\mathcal{D}$(best) | 0.565 | 0.884 | 0.400 | 0.439 | 0.467 | / |
| RaM-MSE (w/ Aug.) | **0.933 ± 0.032** | **0.957 ± 0.013** | **0.507 ± 0.028** | 0.962 ± 0.031 | **0.674 ± 0.044** | **3.7 / 6** |
| RaM-MSE (w/o Aug.) | 0.928 ± 0.022 | 0.944 ± 0.017 | 0.502 ± 0.015 | **0.983 ± 0.012** | 0.652 ± 0.045 | 4.7 / 6 |
| RaM-RankCosine (w/ Aug.) | **0.940 ± 0.028** | **0.951 ± 0.017** | **0.514 ± 0.026** | **0.982 ± 0.012** | **0.675 ± 0.049** | **2.2 / 6** |
| RaM-RankCosine (w/o Aug.) | 0.929 ± 0.019 | 0.944 ± 0.005 | 0.504 ± 0.018 | 0.980 ± 0.016 | 0.654 ± 0.038 | 4.7 / 6 |
| RaM-ListNet (w/ Aug.) | **0.949 ± 0.025** | 0.962 ± 0.015 | **0.517 ± 0.029** | **0.981 ± 0.012** | **0.670 ± 0.035** | **2.0 / 6** |
| RaM-ListNet (w/o Aug.) | 0.938 ± 0.025 | **0.964 ± 0.011** | 0.507 ± 0.007 | 0.975 ± 0.010 | 0.640 ± 0.037 | 3.7 / 6 |

Table 10: Ablation studies on output adaptation, considering learning with MSE, RankCosine, and ListNet, which are the best-performing pointwise, pairwise, and listwise loss, respectively, as shown in Table 8. For each combination of loss and task, the better performance is **Bolded**. $\mathcal{D}$(best) denotes the best score in the offline dataset.

| Method | Ant | D'Kitty | Superconductor | TF-Bind-8 | TF-Bind-10 | Mean Rank |
|---|---|---|---|---|---|---|
| $\mathcal{D}$(best) | 0.565 | 0.884 | 0.400 | 0.439 | 0.467 | / |
| RaM-MSE (w/ Adapt.) | **0.933 ± 0.032** | **0.957 ± 0.013** | **0.507 ± 0.028** | 0.962 ± 0.031 | **0.674 ± 0.044** | **3.8 / 6** |
| RaM-MSE (w/o Adapt.) | 0.913 ± 0.028 | 0.953 ± 0.012 | 0.506 ± 0.024 | **0.966 ± 0.023** | 0.653 ± 0.030 | 5.2 / 6 |
| RaM-RankCosine (w/ Adapt.) | **0.940 ± 0.028** | 0.951 ± 0.017 | **0.514 ± 0.026** | **0.982 ± 0.012** | **0.675 ± 0.049** | **2.7 / 6** |
| RaM-RankCosine (w/o Adapt.) | 0.908 ± 0.023 | **0.955 ± 0.015** | **0.514 ± 0.025** | 0.970 ± 0.016 | 0.649 ± 0.019 | 4.5 / 6 |
| RaM-ListNet (w/ Adapt.) | **0.949 ± 0.025** | **0.962 ± 0.015** | **0.517 ± 0.029** | **0.981 ± 0.012** | **0.670 ± 0.035** | **1.6 / 6** |
| RaM-ListNet (w/o Adapt.) | 0.932 ± 0.034 | 0.961 ± 0.013 | 0.516 ± 0.029 | 0.968 ± 0.016 | 0.655 ± 0.015 | 3.2 / 6 |

## F.4  ABLATION OF THE LIST LENGTH $m$

Note that the list length $m$ in the training data could have a impact on the generalization ability of the model (Lan et al., 2009; Tewari & Chaudhuri, 2015) and its impact on OOD generalization ability of LTR algorithms is undiscovered (Chapelle et al., 2010). Besides, popular benchmarks in LTR (Qin et al., 2010b; Qin & Liu, 2013; Chapelle & Chang, 2011; Dato et al., 2016) have different settings of the list length, ranging from 5 to 1000.

Hence, to meet the settings of different LTR benchmarks and to better understand the sensitivity of RaM-ListNet with respect to $m$, we conduct a careful ablation study of the setting of list length $m$, with values varying in $\{10, 20, 50, 100, 200, 500, 1000, 1500, 2000\}$, as shown in Table 11.

Table 11: 100th percentile normalized score in Design-Bench of RaM-ListNet with varying values of $m$, where the best and runner-up results on each task are **Blue** and **Violet**. $\mathcal{D}$(best) denotes the best score in the offline dataset.

| $m$ | Ant | D'Kitty | Superconductor | TF-Bind-8 | TF-Bind-10 | Mean Rank |
|---|---|---|---|---|---|---|
| $\mathcal{D}$(best) | 0.565 | 0.884 | 0.400 | 0.439 | 0.467 | / |
| 10 | $0.916 \pm 0.014$ | $0.953 \pm 0.016$ | $0.508 \pm 0.017$ | **$0.984 \pm 0.007$** | $0.655 \pm 0.063$ | 6.6 / 9 |
| 20 | $0.913 \pm 0.026$ | $0.954 \pm 0.011$ | $0.518 \pm 0.022$ | $0.972 \pm 0.016$ | $0.670 \pm 0.043$ | 6.2 / 9 |
| 50 | **$0.930 \pm 0.035$** | **$0.963 \pm 0.014$** | $0.524 \pm 0.021$ | $0.978 \pm 0.014$ | $0.653 \pm 0.046$ | **3.5 / 9** |
| 100 | $0.922 \pm 0.024$ | **$0.963 \pm 0.015$** | **$0.525 \pm 0.020$** | $0.967 \pm 0.016$ | **$0.702 \pm 0.129$** | **3.4 / 9** |
| 200 | $0.927 \pm 0.023$ | $0.960 \pm 0.017$ | **$0.526 \pm 0.020$** | $0.977 \pm 0.015$ | $0.650 \pm 0.015$ | 4.6 / 9 |
| 500 | $0.920 \pm 0.028$ | **$0.962 \pm 0.007$** | $0.518 \pm 0.028$ | $0.975 \pm 0.011$ | **$0.699 \pm 0.128$** | 4.0 / 9 |
| 1000 | **$0.949 \pm 0.025$** | **$0.962 \pm 0.015$** | $0.517 \pm 0.029$ | **$0.981 \pm 0.012$** | $0.670 \pm 0.035$ | **3.4 / 9** |
| 1500 | $0.918 \pm 0.030$ | $0.961 \pm 0.012$ | $0.511 \pm 0.025$ | $0.971 \pm 0.019$ | $0.691 \pm 0.127$ | 5.8 / 9 |
| 2000 | $0.905 \pm 0.035$ | $0.958 \pm 0.016$ | $0.515 \pm 0.027$ | $0.967 \pm 0.020$ | $0.664 \pm 0.043$ | 7.5 / 9 |

From the results in Table 11, we observe that RaM-ListNet obtains the best performance when $m = 100$ or $m = 1000$, and it will undergo a score drop as $m$ gets relatively large, which demonstrates the need for a careful tuning of the list length.

### F.5 Results on OOD Metrics

In this subsection, we also present the OOD-MSE results in Table 12 and OOD-AUPCC values in Table 13. As the results deliver, RaM combined with RankCosine or ListNet perform poor in OOD-MSE, while they rank the best two in OOD-AUPCC. Coupled with the fact that RaM obtains the best performance in our main results (Table 2), results on OOD-MSE and OOD-AUPCC further demonstrate that: 1) an algorithm with better OOD-AUPCC could result in better performance in offline MBO, no matter what its OOD-MSE is; 2) ranking loss is more suitable than MSE for offline MBO, since RaM obtains a better OOD-AUPCC compared to other regression-based methods.

Table 12: OOD-MSE of different methods in Design-Bench, where the best and runner-up results on each task are **Blue** and **Violet**.

| Method | Ant | D'Kitty | Superconductor | TF-Bind-8 | TF-Bind-10 | Mean Rank |
|---|---|---|---|---|---|---|
| Grad. Ascent | **$9.134 \pm 0.821$** | $0.444 \pm 0.123$ | $1.054 \pm 0.229$ | $5.543 \pm 0.263$ | $2.930 \pm 0.171$ | 3.6 / 7 |
| COMs | **$9.084 \pm 0.514$** | **$0.303 \pm 0.104$** | **$0.930 \pm 0.043$** | $5.941 \pm 0.201$ | $2.541 \pm 0.047$ | **2.6 / 7** |
| IOM | $9.520 \pm 0.948$ | **$0.299 \pm 0.062$** | **$0.798 \pm 0.041$** | $8.779 \pm 0.364$ | $2.594 \pm 0.069$ | **3.0 / 7** |
| ICT | $160468.083 \pm 354.495$ | $71634.341 \pm 101.262$ | $8444.529 \pm 60.098$ | **$0.163 \pm 0.037$** | **$0.352 \pm 0.060$** | 4.6 / 7 |
| Tri-Mentoring | $160641.446 \pm 111.371$ | $71557.860 \pm 25.979$ | $8190.870 \pm 1.785$ | **$0.115 \pm 0.000$** | **$0.381 \pm 0.030$** | 4.4 / 7 |
| **RaM-RankCosine (Ours)** | $17.113 \pm 0.174$ | $1.503 \pm 0.092$ | $11.170 \pm 0.293$ | $18.799 \pm 1.345$ | $2.632 \pm 0.129$ | 5.2 / 7 |
| **RaM-ListNet (Ours)** | $25.011 \pm 0.824$ | $2.689 \pm 0.249$ | $8.999 \pm 2.449$ | $11.546 \pm 3.295$ | $2.408 \pm 0.315$ | 4.6 / 7 |

Table 13: OOD-AUPCC of different methods in Design-Bench, where the best and runner-up results on each task are **Blue** and **Violet**.

| Method | Ant | D'Kitty | Superconductor | TF-Bind-8 | TF-Bind-10 | Mean Rank |
|---|---|---|---|---|---|---|
| Grad. Ascent | $0.363 \pm 0.028$ | $0.403 \pm 0.002$ | **$0.731 \pm 0.006$** | $0.670 \pm 0.017$ | $0.518 \pm 0.009$ | 4.7 / 7 |
| COMs | **$0.744 \pm 0.015$** | **$0.727 \pm 0.005$** | $0.391 \pm 0.028$ | $0.433 \pm 0.002$ | $0.505 \pm 0.001$ | 4.4 / 7 |
| IOM | **$0.649 \pm 0.044$** | **$0.648 \pm 0.085$** | $0.436 \pm 0.057$ | $0.428 \pm 0.007$ | $0.515 \pm 0.056$ | 4.6 / 7 |
| ICT | $0.443 \pm 0.095$ | $0.549 \pm 0.089$ | $0.596 \pm 0.100$ | $0.658 \pm 0.015$ | $0.545 \pm 0.023$ | 4.6 / 7 |
| Tri-Mentoring | $0.346 \pm 0.000$ | $0.403 \pm 0.000$ | **$0.740 \pm 0.003$** | $0.690 \pm 0.000$ | **$0.571 \pm 0.000$** | 3.7 / 7 |
| **RaM-RankCosine (Ours)** | $0.492 \pm 0.013$ | $0.437 \pm 0.013$ | $0.713 \pm 0.003$ | **$0.714 \pm 0.004$** | $0.551 \pm 0.009$ | **3.2 / 7** |
| **RaM-ListNet (Ours)** | $0.474 \pm 0.045$ | $0.628 \pm 0.017$ | $0.723 \pm 0.004$ | **$0.709 \pm 0.005$** | **$0.562 \pm 0.008$** | **2.8 / 7** |

