# OpenReview forum: "Offline Model-Based Optimization by Learning to Rank"
_ICLR.cc/2025/Conference — ICLR 2025 Poster_

### Official Review · Reviewer_wgm9 · 2024-10-30

**Soundness:** 3
**Presentation:** 3
**Contribution:** 3
**Rating:** 8
**Confidence:** 5

**Summary:**

This paper challenges the use of MSE for training surrogate models in offline MBO, demonstrating its weak correlation with final performance. Instead, the authors propose a ranking-based model leveraging AUPRC, which achieves superior results compared to existing methods and suggests promising new directions for offline MBO.

**Strengths:**

- Utilizing ranking scores in place of MSE to train the offline optimization surrogate model is an innovative approach. It aligns more intuitively with the objectives of offline optimization, making it a fitting metric for this context.

- The algorithm is very clear and easy to understand. At the same time, this paper conducts a theoretical analysis of generalization error bound.

- The authors conducte a wide range of experiments, providing comprehensive comparisons with the latest methods. The experimental results show the feasibility and effectiveness of the proposed approach.

- This article is well-structured, with clear logic and easy-to-follow explanations that enhance readability and comprehension.

- The supplementary material is sufficient and the algorithm can be fully reproduced without any problems. The appendices contain ample additional experiments and ablation studies. The appendices also contain a very comprehensive review of related work.

**Weaknesses:**

- Some concepts need to be clarified, such as OOD in the ranking model and Recall@k in Definition 1. These are raised in the "questions" below.

- Typos. For example, "e.g" in line 48 should be "e.g.", "caculated" in line 241 should be "calculated". A thorough proofreading is recommended.

**Questions:**

1. As mentioned by the authors in the introduction, one of the key challenges in offline optimization is Out-of-Distribution (OOD) data. While this ranking approach offers a better indication than the traditional OOD-MSE and aligns well with the offline optimization context, how do the authors address the OOD issue in the ranking setting? Could you provide a detailed analysis of the OOD problem in this context?

2. In Definition 1, regarding Recall@k, why does the Recall@k value depend solely on k and N, with no relation to the model's predictions or true ranking? Additionally, this definition does not seem to reflect the model’s recall performance. Should the authors consider revising the definition, or reconsider using the term "recall" to refer to this variable?

3. In Table 2, the results for backward methods like BONET[1] and GTG[2] appear to be much lower than those in the original paper (Table 2 in [1] and Table 1 in [2]). Could this be due to differences in experimental settings? If so, why don't the authors use the same configuration?

[1] Generative pretraining for black-box optimization. ICML 2023.

[2] Guided trajectory generation with diffusion models for offline model-based optimization. NeurIPS 2024.

---

> ### Author Response · Authors · 2024-11-18
> **Response by the authors (1/2)**
>
> Thank you for your valuable and encouraging comments. Below please find our responses.
>
> ### Q1 Minor typos
> Thanks for pointing these out. We have revised them and proofread thoroughly according to your suggestions. Thank you very much.
>
> ### Q2 How to address the OOD issue in the ranking setting and provide a detailed analysis?
>
> There have been various efforts to mitigate the OOD issue in offline MBO, but those techniques are done on the regression-based framework, which are not specially designed for the ranking-based framework. However, some of the techniques can also be applied to our method.
> In Table 3, we examined this by simply replacing the MSE loss of prior methods with the ranking loss, i.e., ListNet, which resulted in better performance.
> This shows the possibility to apply prior methods that aim to mitigate OOD issue in offline MBO to our proposed ranking-based framework, and more tricks that target towards model's generalization ability to properly rank samples remain unexplored (e.g., similar to COMs [1], how to design conservative objectives in the ranking setting), which is an interesting future work.
>
> Besides, another important direction to address the OOD issue in the ranking setting is to utilize more complex surrogate modeling.
> In the field of LTR, it is well recognized that a simple multilayer perceptron (MLP) is inadequate to capture the ranking information conveyed by the ranking loss [2-4].
> However in this work, in order to fairly compare to the regression-based methods, we simply parameterize the surrogate model as an MLP. More complex surrogate structures, such as DASALC [4], can be considered for offline MBO.
>
> ### Q3 Revisiting the definition or reconsidering the term ``recall" in Definition 1
>
> Thank you for your valuable suggestions! We acknowledge that the definition of Precision and Recall in Definition 1 is confusing on $\operatorname{Recall}@k$. We apologize for that.
>
> We define the metric *AUPRC for offline MBO* based on the ranking perspective and the need to select top-$k$ designs in offline MBO, where $\operatorname{Precision}@k$ reflects the model's ability to select the top-$k$ designs according to varying $k$, and $\operatorname{Recall}@k$ reflects the coverage of $k$. With abuse of notation, $\operatorname{Recall}@k$ is different from the standard form in information retrieval since we do not have a relevant set in the context of offline MBO.
>
> Following your suggestions, we have renamed $\operatorname{Recall}@k$ to $\operatorname{Coverage}@k$ and changed the term of *AUPRC for offline MBO* to *Area Under the Precision-Coverage Curve (AUPCC) for offline MBO* in our paper to prevent misunderstanding. Thank you again for this suggestion.

---

> ### Author Response · Authors · 2024-11-18
> **Response by the authors (2/2)**
>
> ### Q4 Experimental results for BONET [5] and GTG [6]
>
> Thank you for your comment. We double-checked the experimental setting of these two recent backward methods, and confirm the correctness of our results in Table 2. Detailed information of our settings and reproduction on these two methods are listed below:
>
> - BONET. We use the open-source repository released by the authors of BONET (https://github.com/siddarthk97/bonet) for reproduction. Since the repository does not provide instructions to reproduce, we run BONET based on our understanding on its code. Specifically, we ran ``scripts/<task_name>/train.sh`` for model training and ``scripts/<task_name>/test.sh`` for design sampling. We set the hyper-parameters according to the original paper, except that for the evaluation budget, we changed it from 256 to 128 for a fair comparison (since other methods use a budget of 128). We observed the performance drop after changing the evaluation budget, which is also mentioned in Appendix C of the GTG paper [6]. We also observed that our BONET results in Table 2 are similar to the results in Table 1 of GTG paper. Thus, we believe that our results for BONET are reasonable.
>
> - GTG. We use the open-source repository released by the authors of GTG (https://github.com/dbsxodud-11/GTG) for reproduction. We double-checked the hyper-parameter settings in the ``config/`` folder and found that the settings are correct. Specifically, the hyper-parameters of trajectories construction in Table 8 of the original paper are set in line 99-122 of ``construct_trajectories.py``. We set hyper-parameters, train the model, sample candidate, and evaluate GTG based on the recommended methods in the repository. Taking ``Ant`` task for an example, we run GTG by conducting these commands sequentially:
> ```bash
> python construct_trajectories.py --task ant
> python train.py --task ant --horizon 64 --k 20 --n_traj 4000 --eps 0.05 --seed <seed>
> python evaluate.py --task ant --horizon 64 --k 20 --n_traj 4000 --eps 0.05 --ctx_len 32 --alpha 0.8 --seed <seed>
> ```
> Detailed settings of ``horizon``, ``k``, ``eps``, ``n_traj`` follow Table 8 in the GTG paper, and settings of ``ctx_len`` and ``alpha`` follow Section 4.4 in the paper. We checked the hyper-parameters again and found that our results are correct. The difference between our results and the original results might result from the difference of machines or hardware, because we use RTX 4090 GPU while they use RTX 3090 GPU.
>
> We ran these two methods using their open-source code, following either their provided instructions or our understanding. Please let us know if we made any mistakes in the execution process.
>
> **We hope that our response has addressed your concerns, but if we missed anything please let us know.**
>
> ### References
>
> [1] Brandon Trabucco, Aviral Kumar, Xinyang Geng, Sergey Levine. Conservative objective models for effective offline model-based optimization. In Proceedings of the 38th International Conference on Machine Learning (ICML'21), pp. 10358-10368, Virtual, 2021.
>
> [2] Rama Kumar Pasumarthi, Honglei Zhuang, Xuanhui Wang, Michael Bendersky, Marc Najork. Permutation equivariant document interaction network for neural learning-to-rank. In Proceedings of the 2020 ACM SIGIR International Conference on the Theory of Information Retrieval (ICTIR'20), pp.145-148, Virtual, 2020.
>
> [3] Przemyslaw Pobrotyn, Tomasz Bartczak, Mikolaj Synowiec, Radoslaw Bialobrzeski, Jaroslaw Bojar. Context-aware learning to rank with self-attention. In Workshop on eCommerce @ SIGIR'20, Virtual, 2020.
>
> [4] Zhen Qin, Le Yan, Honglei Zhuang, Yi Tay, Rama Kumar Pasumarthi, Xuanhui Wang, Mike Bendersky, Marc Najork. Are neural rankers still outperformed by gradient boosted decision trees? In Proceedings of the 9th International Conference on Learning Representations (ICLR), Virtual, 2021.
>
> [5] Siddarth Krishnamoorthy, Satvik Mehul Mashkaria, Aditya Grover. Generative pretraining for black-box optimization. In Proceedings of the 40th International Conference on Machine Learning (ICML'23), pp. 24173–24197, Honolulu, HI, 2023.
>
> [6] Taeyoung Yun, Sujin Yun, Jaewoo Lee, Jinkyoo Park. Guided trajectory generation with diffusion models for offline model-based optimization. In Advances in Neural Information Processing Systems 37 (NeurIPS'24), Vancouver, Canada, 2024.

---

> > ### Comment · Reviewer_wgm9 · 2024-11-21
> > **Reply to the authors**
> >
> > Thank you for your prompt reply! Your answer solved my questions and I no longer have any concerns.
> >
> > This paper proposes a novel method that uses a ranking-based model to deal with offline optimization problems. In this paper, the authors analyze the method in detail, design AUPRC and conduct a theoretical analysis of the Generalization Error Bound. The authors also design sufficient and detailed experiments to illustrate the superiority and effectiveness of the method.
> >
> > I have improved my score and will recommend this article to accept!

---

> > > ### Author Response · Authors · 2024-11-21
> > > **Thanks for your feedback!**
> > >
> > > Thanks for your feedback! We are glad to hear that your concerns have been addressed. We sincerely appreciate the time and effort you have dedicated to reviewing our paper and providing thoughtful and valuable comments.
> > >
> > > Best regards
> > >
> > > Authors

---

### Official Review · Reviewer_ChpN · 2024-11-01

**Soundness:** 3
**Presentation:** 4
**Contribution:** 3
**Rating:** 6
**Confidence:** 4

**Summary:**

This paper proposes a new approach for offline optimization which focuses on training a surrogate to rank samples rather than to accurately predict their values. This surrogate is trained using the LTR framework with ListNet loss.

**Strengths:**

Paper was clearly written and original. To the best of my knowledge, the LTR framework was not previously used for offline optimization. Empirical results show positive improvement over previous baselines.

**Weaknesses:**

I think the idea that MSE is not always optimal for offline opt was recently pointed out in the MATCH-OPT paper (Hoang et al., 2024) (in which they use the term "value-matching surrogates" to describe model trained with MSE loss). Personally, I think that MSE is not worse than any other metric for optimization, but it could be harder to be estimated accurately for OOD samples. This seems to have been acknowledged in the MATCH-OPT paper. Section 3.1 and 3.2 of this paper seems to focus on justifying this insight using empirical evidence. 3.3 describes the LTR algorithm (which is not original), and 3.4 quotes some generalization result of LTR.

Regarding 3.1 and 3.2, it makes no sense to compare the two metrics MSE and AUPRC. MSE evaluation does not use info about the top-k design (which has very strong correlation with the rank of the 100th percentile candidate), whereas AUPRC explicitly uses this info. It seems obvious to me that the latter would have higher Pearson correlation. Additionally, one cannot use the top-k info during training, so this would have little relevance to the actual method (which uses the ListNet loss). Most offline opt frameworks don't simply rely on the MSE loss though. As pointed out in the intro, there are some that use generative modeling or regularization to condition the surrogate model. How do those techniques relate to the idea of ranking samples?

The theory is, unfortunately, quite shallow and needs some extension. Theorem 1 (which is more of a proposition), states an equivalence, so the reverse argument could also be put forward -- why not MSE instead of ranking? As mentioned, I think the key challenge of offline opt is that we cannot control the surrogate's behavior OOD. Early error can lead the gradient search astray, so one should focus on quantifying this compounding effect.

Theorem 2 is a direct quote from previous work, but it seems to add no value to the current setting. Does LTR generalize better than MSE on OOD data? I doubt that we can conclude anything general like that without a particular set of assumptions. So, the right question is, in which scenario will LTR generalize better than MSE on OOD data? I think this is important to answer this, especially when this paper is trying to advocate for a paradigm shift in offline opt.

Empirical results seem reasonable. However, given the landscape of offline opt research, in which every recent paper seems to report a different set of results on the Design Bench (typically suggesting that it is best), I find it hard to conclude that any method is more significant and robust than the others without proper theoretical backing. The paper should also compare with the recent DDOM method (Krisnamoorthy et al., 2023) which is representative of the generative modeling line of work.

**Questions:**

Please see above.

**Details Of Ethics Concerns:**

No ethical concerns.

----
I have read the rebuttal and adjusted the rating accordingly.

---

> ### Author Response · Authors · 2024-11-18
> **Response by the authors (1/4)**
>
> Thank you for your valuable and constructive comments. Below please find our responses.
>
> ### Q1 The idea that MSE is not always optimal for offline opt was recently pointed out in the MATCH-OPT paper
>
> Thanks for pointing out this important observation! MATCH-OPT [1] has mentioned that focusing on value matching alone is not suitable for offline MBO. In this paper, we conducted a thorough and comprehensive analysis, providing empirical evidence that OOD-MSE is not closely related to the final design quality in offline MBO (as shown in Figure 2 and Table 1) and demonstrating the significance of training an order-preserving surrogate model (in Theorem 1).
>
> We have updated our paper to highlight this important observation of MATCH-OPT in our Introduction, Section 2.1, and Appendix A, with the changes colored in red. Thank you again for your valuable comments.
>
> ### Q2 MSE is not worse than any other metric for optimization, but it could be harder to be estimated accurately for OOD samples
>
> Good point! We agree that MSE is not bad for optimization, since a model that can predict well on OOD region is quite good to optimize inside it. However, prior works have pointed out that OOD-MSE is naturally difficult to learn [2-3]. Recent works like MATCH-OPT [1] and Tri-Mentoring [4] focus on including more information (e.g., gradient information and weak pairwise supervision signal), but these works are based on the regression-based framework trained with MSE. In this work, we thoroughly propose a novel framework based on ranking, which is a weaker condition since it only commands for correct orders, while minimizing MSE argues for both order preserving and value approximating. In Theorem 1, we have demonstrated that an order-preserving surrogate model is sufficient for offline MBO by stating the equivalence. We examined that ranking metric is more efficient than MSE for offline MBO from an empirical perspective in Figure 2 and Table 1. The experimental results also show the strengths of our framework.
>
> Besides, in Table 3, we combined our framework with approaches in offline MBO by simply replacing the MSE term in their loss with ListNet and incorporating output adaptation, to further show the effectiveness of the ranking framework. Here we also provide new experimental results on MATCH-OPT + ListNet and original MATCH-OPT, consistent with our observation in Table 3.
>
>
> | Method    | Type    | Ant           |       | D'Kitty       |       | Superconductor |       | TF-Bind-8     |       | TF-Bind-10    |       |
> |-----------|---------|---------------|-------|---------------|-------|----------------|-------|---------------|-------|---------------|-------|
> |           |         | Score         | Gain  | Score         | Gain  | Score          | Gain  | Score         | Gain  | Score         | Gain  |
> | MATCH-OPT | MSE     | 0.933 ± 0.016 |       | 0.952 ± 0.008 |       | 0.504 ± 0.021  |       | 0.824 ± 0.067 |       | 0.655 ± 0.050 |       |
> |           | ListNet | 0.936 ± 0.027 | +0.3\% | 0.956 ± 0.018 | +0.4\% | 0.513 ± 0.011  | +1.8\% | 0.829 ± 0.089 | +0.6\% | 0.659 ± 0.037 | +0.6\% |
>
> We have revised to add these new experimental results to Table 3. Thank you.
>
> ### Q3 It makes no sense to compare the two metrics MSE and AUPCC (i.e., AUPRC in the primitive version)
>
> Thank you for your comment. As mentioned above, the ability to preserve order is sufficient for surrogate models in offline MBO, which is a weaker requirement than minimizing MSE. Given that AUPCC is a ranking-related metric and reflects how well the model identifies the top-$k$ designs, AUPCC can intuitively be better than MSE. As this is the first work to explicitly propose a ranking-based framework for offline MBO, we conducted a thorough and systematic analysis with empirical evidence and theoretical statement.
>
> ### Q4 The top-$k$ info cannot be used during training, which is of little relevance to the method.
>
> Thank you for your comment, but there may be misunderstandings. In this work, we do not just simply replace MSE loss with LTR loss (e.g., ListNet). Note that the ranking information (e.g., top-$k$ info) is limited in the primitive offline dataset with unordered $(\mathbf{x},y)$ pairs. For better ranking-based model learning, we apply data augmentation to construct a dataset to obtain and utilize the ranking information, where each entry is a list of items to be sorted. Specifically, as illustrated in line 3-7 in Algorithm 1, we first randomly sample $m$ design-score pairs from the offline dataset to form a list to be sorted, then repeat this for $n$ times to construct the augmented dataset with $n$ lists. With this technique, we can use ListNet loss to feed the ranking information (e.g., top-$k$ info) to the surrogate model training process. These details can be found in Section 3.3. Thus, we believe that our method is capable to leverage such information to train a better surrogate model.

---

> ### Author Response · Authors · 2024-11-18
> **Response by the authors (2/4)**
>
> ### Q5 How do prior techniques relate to the idea of ranking samples?
>
> Thank you for your valuable comment. There are various approaches in the context of offline MBO, and here we introduce some that may relate to the idea of ranking samples from our comprehension:
>
> - MATCH-OPT [1]. MATCH-OPT claims that value matching is not enough for offline MBO, and enforces the model to match the ground-truth gradient, which is approximated via interpolation on the constructed trajectories. The idea of matching gradient is related to ranking samples since a model with proper gradient could reflect the relationship in a small neighborhood.
>
> - Tri-Mentoring [4]. Tri-Mentoring facilitates mentoring among three parallel proxies through a combination of a voting-based pairwise supervision module and an adaptive soft-labeling module, resulting in a more robust ensemble of proxies. Each proxy uses weak semi-supervised pairwise-ranking-based voting signals provided by other proxies to fix its predictions and finetune its weights.
>
> - BONET [5]. BONET constructs monotonic trajectories to mimic the behavior of a black-box optimizer and uses them to train an autoregressive model, after which the final designs can be sampled under a regret-to-go heuristic. Here the trajectories are constructed by ranking the collected samples, from which the model may capture some ranking information.
>
> In this work, we directly identify the idea of ranking samples, and conduct a systematic analysis on this view. After that, we reformulate the objective of the training process by replacing the core MSE loss with a ranking loss, and apply data augmentation and output adaptation for model training and solution search, respectively. Thanks to your suggestion, we have revised to add some discussion on the relationship between prior methods and ranking in Appendix A.
>
> Besides, our proposed ranking-based framework and these approaches are not mutually exclusive; rather, our framework can be combined with such approaches to enhance model's generalization ability. In Table 3 and our experiment in the reply to Q2, we examine this by simply replacing the MSE loss with ListNet and incorporating output adaptation, which demonstrates the versatility of ranking loss and the possibility to apply prior methods in offline MBO to our proposed ranking framework.

---

> ### Author Response · Authors · 2024-11-18
> **Response by the authors (3/4)**
>
> ### Q6 The theory needs extensions
>
> Thanks for your constructive suggestion. As mentioned in our reply to Q2, learning to rank samples correctly is a weaker condition than learning to minimize MSE, since minimizing MSE commands for both order preserving and value matching. The aim of proving the equivalence in Theorem 1 is to show that the weaker condition, order preserving, is sufficient, which motivates the proposal of directly learn the ranking information by leveraging LTR techniques.
>
> As for the question you raised, ``in which scenario will LTR generalize better than MSE on OOD data?", we are also curious about the question when conducting this work. We tried to identify some theoretical support or evidence from prior works, but surprisingly we cannot find that even in the field of LTR.
>
> We also attempted to analyze this point theoretically, but encountered several challenges. Below we briefly present the most promising approach we explored and the difficulties we face.
>
> - Try to find a special function class $\mathcal{F}$, from which the ranking model $\hat{f}$ to be learned is, such that models learned with LTR techniques have an upper bound guarantee on some ranking measure while models trained with MSE do not. Formally, let $R$ be a ranking measure (which can be the expected risk of a specific ranking loss or a ranking metric, e.g., NDCG), and denote the empirical risk of model trained with LTR and that trained with MSE as $\hat{R} _ {LTR}$ and $\hat{R} _ {MSE}$, respectively. For ease of exposition, $R$ here refers to the expected risk in Theorem 2. From Theorem 2, $R$ can be upper bounded by $\hat{R} _ {LTR}$ as
> $$
> \sup (R - \hat{R} _ {LTR}) \leq \frac{C _ {\mathcal{A}} (\phi) N(\phi) }{\sqrt{n}} + \sqrt{\frac{2\ln (2/\delta)}{n}}.
> $$
> Then, if we could find that for $\mathcal{F}$, $R - \hat{R} _ {MSE}$ always has a slower convergence rate, i.e.,
> $$
> R - \hat{R} _ {MSE} \geq \mathcal{O}(\frac{1}{\sqrt{n}}),
> $$
> we can show that models learned with MSE are worse than that learned with LTR. However, such an analysis can be difficult because: 1) There is no theoretical evidence to show the generalization bound on ranking by optimizing MSE. 2) Most generalization bound analysis in LTR assume i.i.d (as Theorem 2 in our paper), while OOD analysis in LTR is quite limited.
>
> - Identify a special case that supports this intuition.
> Assume that the function class $\mathcal{F}$ is a linear function class and the offline data is drawn from a ground-truth function $f$ with long-tailed noise on the objective value. Models trained with MSE are susceptible to heavy-tailed noise, as the mean of $y$ is heavily influenced in regions with such noise. In contrast, models trained with pairwise ranking loss demonstrate greater stability in such scenarios. An illustrative example could be as follows. Assume that the ground-truth function is $f(\mathbf{x}) = \mathbf{x}^2$ and the offline dataset $\mathcal{D}=$ { $(1,1), (1.9, 3.7), (2.1, 4.5), (2, -12)$ } where $(2, -12)$ suffers from the heavy-tailed noise. Then, the model trained with MSE would exhibit negative correlation, while that trained with LTR would demonstrate positive correlation, which shows that the model trained with LTR is more robust. However, such counterexamples are still based on strong assumptions. A well-constructed example with theoretical support remains unexplored.
>
>
> Above are the possible approaches that we have attempted from the theoretical view. Thank you for pointing this out, and we have revised to discuss the probable approaches and difficulties for theoretical analysis in Appendix C. We also welcome the reviewers to discuss on the topic for this emerging field.
>
>
> ### Q7 Hard to conclude from results on offline optimization papers that any method is more significant and robust
>
> Thanks for your valuable comment. We also notice this issue. To make the results in our paper reproducible, we use the open-source code provided by each paper and set hyper-parameters carefully according to the description in their papers. We have mentioned this in Appendix E.4, and listed the links of all open-source repositories we used.
>
> Besides, in order to help the community better reproduce the results, we will open-source not only all the codes but also the model weights in our paper, as we mentioned in the ``readme.md`` file of our supplemental files.
>
> ### Q8 Lack of comparison to the DDOM method [6]
> We compared DDOM in line 11 of Table 2 in our submitted version. We believe that we have done a comprehensive comparison to other 20 methods. These experimental results clearly demonstrate the superior performance of our ranking-based model over regression-based model.
>
> **We hope that our response has addressed your concerns, but if we missed anything please let us know.**

---

> > ### Comment · Reviewer_ChpN · 2024-11-25
> > **Thanks for your response**
> >
> > My apology for overlooking the DDOM results. I also appreciate the added comparison to MATCH-OPT and the discussion on prior techniques & am willing to raise my score to 6. As for the rest of the points, I don't really think we disagree too much. What I meant by "Additionally, one cannot use the top-k info during training, so this would have little relevance to the actual method" -- is that AUPRC or AUPPC are not the objective in training, so the preliminary result section (3.1/3.2) has little relevance to the actual method, which uses ListNet loss. I'm aware of the data augmentation part of your algorithm.
> >
> > Regarding potential approaches for the theory -- I think the general direction makes sense and wish you the best in deriving something interesting.

---

> > > ### Author Response · Authors · 2024-11-26
> > > **Thanks for your feedback**
> > >
> > > Thanks for your feedback! We are glad to hear that your concerns have been addressed. We sincerely appreciate the time and effort you have dedicated to reviewing our paper and providing thoughtful and valuable comments.
> > >
> > > Best regards
> > >
> > > Authors

---

> ### Author Response · Authors · 2024-11-18
> **Response by the authors (4/4)**
>
> ### References
>
> [1] Minh Hoang, Azza Fadhel, Aryan Deshwal, Jana Doppa, Trong Nghia Hoang. Learning surrogates for offline black-box optimization via gradient matching. In Proceedings of the 41st International Conference on Machine Learning (ICML'24), pp. 18374-18393, Vienna, Austria, 2024.
>
> [2] Brandon Trabucco, Xinyang Geng, Aviral Kumar, and Sergey Levine. Design-Bench: Benchmarks for data-driven offline model-based optimization. In Proceedings of the 39th International Conference on Machine Learning (ICML'22), pp. 21658–21676, Baltimore, MD, 2022.
>
> [3] Brandon Trabucco, Aviral Kumar, Xinyang Geng, Sergey Levine. Conservative objective models for effective offline model-based optimization. In Proceedings of the 38th International Conference on Machine Learning (ICML'21), pp. 10358-10368, Virtual, 2021.
>
> [4] Can (Sam) Chen, Christopher Beckham, Zixuan Liu, Xue (Steve) Liu, Chris Pal. Parallel-mentoring for offline model-based optimization. In Advances in Neural Information Processing Systems 36 (NeurIPS'23), pp. 76619-76636, New Orleans, LA, 2023.
>
> [5] Siddarth Krishnamoorthy, Satvik Mehul Mashkaria, Aditya Grover. Generative pretraining for black-box optimization. In Proceedings of the 40th International Conference on Machine Learning (ICML'23), pp. 24173–24197, Honolulu, HI, 2023.
>
> [6] Siddarth Krishnamoorthy, Satvik Mehul Mashkaria, Aditya Grover. Diffusion models for black-box optimization. In Proceedings of the 40th International Conference on Machine Learning (ICML'23), pp. 17842-17857, Honolulu, HI, 2023.

---

> ### Author Response · Authors · 2024-11-25
> **Looking forward to your feedback**
>
> Dear Reviewer ChpN,
>
> Thank you very much for taking the time to carefully review our work and provide your constructive comments. Your feedback has been invaluable in helping us improve our work.
>
> In response to your comments, we have included the following content:
>
> 1. We have updated our paper to highlight the observation of MATCH-OPT. We also combined MATCH-OPT into our framework and conducted experiments in our response to Q2. The experimental results show further improvement in incorporating MATCH-OPT into our proposed ranking framework.
> 2. We have introduced some prior techniques that implicitly relate to the idea of ranking samples or utilize the ranking information in Appendix A.1 and our reply to Q5, while our work directly identifies this point and conducts a more systematic analysis.
> 3. We have discussed the approaches for further theoretical analysis and given a theoretical example in Appendix C.1 and our response to Q6, where LTR losses are superior to MSE in obtaining more robust performance.
>
> Besides, as suggested by Reviewer 9jkq, to better understand the theoretical example, we additionally conducted numerical experiments in Appendix C.2 and our response to Reviewer 9jkq. These results further support our theoretical findings.
>
> May we kindly ask if our response has addressed the issues you raised? If you still have any further questions or concerns, we would be more than happy to continue the discussion.
>
> Thank you again for your time and effort.
>
> Best regards,
>
> Authors

---

### Official Review · Reviewer_9jkq · 2024-11-02

**Soundness:** 3
**Presentation:** 3
**Contribution:** 3
**Rating:** 6
**Confidence:** 4

**Summary:**

Offline model-based optimization (MBO) is concerned with the goal of maximizing an unknown objective function using an offline dataset.
$\min_{x \in X} f(x)$ with unknown $f(.)$. The goal is to learn a proxy function $\hat{f} (.)$.
A dataset from past observations is available $[  (x_i, f(x_i) ]_{i=1}^n$. The training has to be done completely offline, because online evaluation is not possible in this setup.
The rudimentary approach is to train $\hat{f} (.)$ by minimizing the MSE loss between $\hat{f} (x_i)$ and $f (x_i)$. This vanilla approach has the problem of out-of-distribution issue, because $\hat{f} (.)$ does not approximate well on unseen space within $X$.
This paper proposes to minimize a learning-to-rank (LTR) loss instead of the MSE loss and experimentally compare the proposed RankCosine and ListNet losses with the existing approaches.

**Strengths:**

**Originality**: This paper utilizes the existing LTR loss functions for offline MBO problems. This is a novel way to approach the offline MBO problem.

**Quality**: I found some concerns which I will raise in the weakness section.

**Clarity**: Overall, the paper has clarity.

**Significance**: This new approach to offline MBO could be of significance, if the following questions are answered.

**Weaknesses:**

Although the approach is novel, there are concerns regarding how it can generalize performance and alleviate the out-of-distribution issue. It is not convincing that just plugging an LTR loss would address would solve the problem.
(See Questions)

**Questions:**

- In MBO problems, the exact value of the objective function is often critical. For example, when optimizing GPU performance, the goal is not only to minimize latency but also to know the precise latency value to assess the suitability of the design. Using LTR losses, however, does not guarantee that the output will match the true objective. Therefore, a potential drawback of the proposed approach is its failure to accurately estimate the objective function, which is highly relevant in real-world applications.
- I am a bit puzzled by the definitions of Precision and Recall in Definition 1. In machine learning, Precision typically measures the ratio of true positives to all instances classified as positive, while Recall measures the ratio of true positives to the total number of ground-truth positives.  If a design is included in the top-k set, it should be considered a true positive. This suggests the numerator should be the same for both Precision and Recall, the dimension of the intersection and the denominator for both Precision and Recall should be k.
- The out-of-distribution issue in MBO has been well studied. Adversarial weight perturbation [1] is one approach. Perturbing and minimizing the worst case performance among the performance is crucial to have better out-of-distribution issue and to improve generalization. Do you claim that the proposed approach do not require such perturbations?
- To achieve generalization, methods like regularization, ensembling weak learners, and training under adversarial inputs are commonly used. It’s unclear how using an LTR loss alone could eliminate the need for such approaches. Even when training with an LTR loss on a static offline dataset, how you can guarantee that the ranking will generalize to out-of-distribution data.

[1]: Yu, Tianhe, et al. "Combo: Conservative offline model-based policy optimization." Advances in neural information processing systems 34 (2021): 28954-28967.

---

> ### Author Response · Authors · 2024-11-18
> **Response by the authors (1/3)**
>
> Thank you for your valuable comments. Below please find our responses.
>
>
> ### Q1 Probable failure to accurately estimate the objective function
>
> Thank you for your comments. We want to make some clarifications about the goal of offline model-based optimization (MBO). Given a metric $f(\cdot)$ that need to be optimized (e.g., GPU latency you mentioned), offline MBO aims to find the optimal design $\mathbf{x}^* = \mathop{\arg\max}_\mathbf{x} f(\mathbf{x})$ using only the static dataset. This requires the offline MBO algorithm to **recommend $k$ promising designs** for a single batch of true evaluation.
>
> The primary aim of the offline MBO algorithm is to recommend designs with the potential to achieve optimal metrics, which are then sent to true evaluations to ascertain specific performance metrics. Consequently, during the training phase, it is not essential to know the exact metric values.
>
> The prevalent approach for offline MBO is the forward approach, which constructs a surrogate model with $\mathbf{x}$ as input and predicts output values. Traditional methods train this surrogate model using a regression framework based on mean squared error (MSE). However, this approach suffers from out-of-distribution (OOD) issues, which misaligns with the goal of offline MBO, as depicted in Figure 1. We have conducted a comprehensive analysis of this issue and proposed a ranking-based framework to mitigate the OOD issue.
>
> In scenarios where both high-performing designs and their precise scores are critical, our methods may perform less effectively compared to the regression-based models. However, to the best of our knowledge, all offline MBO tasks primarily require recommending suitable candidates, focusing on whether a design is good enough rather than its exact metric value. Therefore, we believe our method is both effective and appropriate for offline MBO tasks.
>
> ### Q2 Issue in definition of Precision and Recall in Definition 1
>
> Thank you for your valuable comments! In the standard definition of AUPRC in information retrieval, a relevant set is needed to indicate the true samples. Since we cannot define a relevant set in the context of offline MBO, we use $\mathcal{D}_0$ as the relevant set for simplification in this work. This simplification distorts the meaning of $\operatorname{Recall}@k$. We sincerely apologize for the caused confusion.
>
> We have revised this part in our paper and hope it is clear now. Here we also give a detailed explanation. In the original paper, we defined the metric *AUPRC for offline MBO* based on the ranking perspective and the need to select top-$k$ designs in offline MBO, where $\operatorname{Precision}@k$ reflects the model's ability to select the top-$k$ designs according to varying $k$, and $\operatorname{Recall}@k$ reflects the coverage of $k$. With abuse of notation, $\operatorname{Recall}@k$ is different from the standard form in information retrieval since we do not have a relevant set in the context of offline MBO. Also according to the suggestion of Reviewer wgm9, we rename $\operatorname{Recall}@k$ as $\operatorname{Coverage}@k$ and change the term of *AUPRC for offline MBO* to *Area Under the Precision-Coverage Curve (AUPCC) for offline MBO* to prevent misunderstanding, which can be found in Definition 1 of our new version of PDF.
>
> We also add discussion on the relationship between AUPCC and ranking in Section 3.2, and that is why we directly utilize LTR techniques for surrogate training.
>
> We sincerely hope that this change could address your concerns. Thank you very much.

---

> > ### Comment · Reviewer_9jkq · 2024-11-21
> > **Reply to Authors**
> >
> > Thank you for the reply. Your response clarifies that the scope of the paper is to recommend designs from a set of designs and not concerned about the exact value.
> > I also agree with the idea of renaming Recall@k as Coverage@k.

---

> > > ### Author Response · Authors · 2024-11-21
> > > **Thank you for your feedback!**
> > >
> > > Dear Reviewer,
> > >
> > > Thank you for your kind response. We are very pleased to hear that our reply has clarified the scope of the paper and addressed your concerns.
> > >
> > > In addition to clarifying the scope, we hope our detailed responses to your other questions (Q3, Q4, and Q5) have adequately addressed your concerns. Specifically:
> > >
> > > - For Q3, we explained that while our ranking-based framework doesn't inherently eliminate the need for generalization tricks like adversarial weight perturbation, it demonstrates the versatility and compatibility of our ranking framework by combining with prior approaches in offline MBO (e.g., Tri-Mentoring, PGS, and MATCH-OPT).
> > > - Regarding Q4, we showed that combining our ranking framework with recent methods led to superior performance, highlighting our approach's potential to integrate effectively with existing strategies.
> > > - For Q5, we emphasized that while the OOD issue is inevitable due to limited offline dataset coverage, our ranking-based framework has shown better generalization in OOD regions compared to regression-based methods, as supported by the OOD-AUPCC results in Table 11.
> > >
> > > We hope these clarifications have resolved your concerns and provided a better understanding of our proposed framework's contributions and potential. If you have any additional questions or feedback, please feel free to let us know.
> > >
> > > Additionally, we hope that with this clearer understanding of the scope and contributions of the paper, you might consider if there is room for a positive adjustment in the evaluation. We would be truly grateful for your consideration.
> > >
> > > Thank you again for your thoughtful comments and the time you have dedicated to reviewing our work.
> > >
> > > Best regards
> > >
> > > Authors

---

> > > > ### Comment · Reviewer_9jkq · 2024-11-21
> > > > **Response**
> > > >
> > > > Thank you again for your prompt response.
> > > > Theorem 2 provides an error bound for LTR losses. Since you are advocating for LTR losses over MSE losses, could you comment on what the error bound for MSE losses would be? Would LTR losses result in a tighter bound? Demonstrating this could strengthen the argument for the applicability of LTR losses over MSE and help convince readers of their advantages.

---

> > > > > ### Author Response · Authors · 2024-11-21
> > > > > **Thank you for your valuable suggestions!**
> > > > >
> > > > > Thank you for your valuable suggestions! We fully recognize the importance of demonstrating the theoretical superiority of LTR losses over MSE. In fact, we actively work on either identifying relevant theoretical foundations from prior works or deriving them by ourselves when conducting this work. Unfortunately, we cannot find such theoretical support even in the field of LTR. When conducting the analysis by ourselves, we also encountered several challenges. Below we briefly outline our most promising investigated approach and discuss the associated challenges.
> > > > >
> > > > > - Find a function class $\mathcal{F}$, from which the ranking model $\hat{f}$ to be learned is, such that models learned with LTR techniques have a tighter upper bound on some ranking measures than models learned with MSE. Here, the term ``ranking measures" can be the expected risk of a specific ranking loss (e.g., $R_{l_{\mathcal{A}}} (f)$ in Theorem 2) or a ranking metric (e.g., NDCG from LTR). Formally, we let the ranking measure $R$ be the expected risk in Theorem 2, and denote the empirical risk of models trained with LTR and that trained with MSE as $\hat{R} _ {LTR}$ and $\hat{R} _ {MSE}$, respectively. From Theorem 2, $R-\hat{R}_{LTR}$ can be upper bounded by
> > > > >
> > > > >     $$
> > > > >     \sup (R - \hat{R} _ {LTR}) \leq \frac{C_{\mathcal{A}}(\phi)N(\phi)}{\sqrt{n}} + \sqrt{\frac{2\ln (2/\delta)}{n}}.
> > > > >     $$
> > > > >
> > > > >     Then, once we find that for any function in $\mathcal{F}$, $R-\hat{R}_{MSE}$ convergences slower, i.e.,
> > > > >
> > > > >     $$
> > > > >     R - \hat{R} _ {MSE} \geq \mathcal{O}(\frac{1}{\sqrt{n}}),
> > > > >     $$
> > > > >
> > > > >     we can conclude that models learned with LTR techniques generalize better than those learned with MSE. However, this can be challenging because: 1) There is no theoretical evidence to show the generalization bound on ranking by optimizing MSE (even in i.i.d scenarios). 2) Most generalization bound analysis in LTR assume i.i.d (as Theorem 2 in our paper), while OOD analysis in LTR is quite limited.
> > > > >
> > > > > - However, we find a counterexample that supports the mentioned theoretical statement we want to make. Let $\mathcal{F}$ be a linear function class and the offline data is collected from a ground-truth function $f$ with long-tailed noise on the objective value. In such scenarios, models trained with MSE are susceptible to heavy-tailed noise, since they depend significantly on the mean value of the objective value over a region, while those trained with pairwise ranking loss have greater robustness. We also provide an illustrative example where the ground-truth function is $f(\mathbf{x})=\mathbf{x}^2$ and the offline dataset $\mathcal{D}=$ { $(1,1),(1.9,3.7),(2.1,4.5),(2,-12)$ }. Here, $(2,-12)$ suffers from the heavy-tailed noise. Then the linear model learned with MSE would exhibit a negative correlation, while that trained with pairwise ranking loss would show a positive correlation, which demonstrates that the model trained with LTR is more stable. However, this special case is still based on strong assumptions. A well-constructed example with theoretical support remains unexplored.
> > > > >
> > > > > Above are promising approaches that we have tried to figure out from the theoretical view. We have discussed this topic in Appendix C of our revised version. Further theoretical analysis is an interesting future work. We also welcome the reviewers to discuss this topic.
> > > > >
> > > > > Although addressing this point theoretically is challenging, LTR can be intuitively more suitable than MSE in the context of offline MBO. As our reply to Q3, learning to rank samples properly is a weaker condition since it only commands for correct orders while minimizing MSE argues for both order preserving and value approximating, and in Theorem 1, we have demonstrated that an order-preserving surrogate model is adequate by showing the equivalence. Besides, in Table 11, we have also shown that using LTR techniques is more effective in generalizing well on OOD ranking performance than regression-based methods by obtaining better OOD-AUPCC results. Thus, we believe the applicability of LTR losses in offline MBO is expected.
> > > > >
> > > > > **We hope that our response has addressed your concerns, but if we missed anything please let us know. Thank you again for your timely response.**

---

> > > > > > ### Comment · Reviewer_9jkq · 2024-11-22
> > > > > > **Response**
> > > > > >
> > > > > > Thank you for the clarification. This discussion in Appendix C is a valuable addition.  Based on the discussion, I am willing to increase the my rating.
> > > > > >
> > > > > > Additionally, I was wondering if the example you mentioned could be extended into a computational experiment. For instance, you could sample instances of $(x, y)$ and introduce heavy-tailed noise to $y$. Then, by comparing the fitted lines using MSE and LTR losses, you could  demonstrate the advantage of LTR losses. Such a simulation could motivate readers and strengthen the case for LTR losses.

---

> > > > > > > ### Author Response · Authors · 2024-11-23
> > > > > > > **Thanks and response (1/2)**
> > > > > > >
> > > > > > > Thanks for your constructive and encouraging suggestions! We sincerely agree that a computational simulation experiment is significant for readers to understand. According to your suggestions, we have conducted additional experiments to strengthen the discussion of the special case and have added them to Appendix C. Below please find the detailed experimental settings and analysis of the results.
> > > > > > >
> > > > > > > **Experimental settings.** Given a dataset $\mathcal{D}=$ { $(x_i, y_i)$ } ${} _ {i=1} ^N$, we want to train a surrogate model $\hat{f}(x)=wx+b$ based on two loss functions, i.e., MSE and RankCosine (a pairwise ranking loss). Details of how to obtain the linear model trained with these two losses are as follows:
> > > > > > >
> > > > > > > - **MSE.** The linear model trained with MSE has a closed-formed solution using the Least Squares Method. Formally, let an augmented matrix $X = [\mathbf{1}, [x_1, x_2, \cdots, x_N]^\top]$, $\mathbf{y} = [y_1,y_2, \cdots, y_N]^\top$ , and $\theta=[w,b]^\top$, and we can obtain that $\theta=(X^\top X)^{-1} X^\top \mathbf{y}$.
> > > > > > > - **RankCosine**: There is no closed-formed solution due to non-linear operations (e.g., vector normalization operator when calculating RankCosine). Hence, we use the Adam optimizer with a learning rate $1\times 10^{-3}$ to search for the optimal value for $w$ and $b$ over 1000 epochs.
> > > > > > >
> > > > > > > We set the ground-truth function to be a quadratic function $f(x)=x^2$ for ease of demonstration, which is increasing and requires $\hat{f}$ having a positive $w$. We assume that the training data is drawn from $[0, 3]$ for better visualization. As for the noise, we initiate the heavy-tailed noises from a Student's t-distribution $g(t) = \frac{\Gamma(\frac{\nu+1}{2})}{\sqrt{\nu\pi}\,\Gamma(\frac{\nu}{2})} \left(1+\frac{t^2}{\nu}\right)^{-\frac{\nu+1}{2}}$ with the degrees of freedom $\nu=2$, and change their magnitude controlled by a scale $\alpha=15$. Besides, to influence the increasing trend, we assume that the heavy-tailed noise is positive for points with $x\in[0, 1.5]$ and negative for points with $x\in(1.5,3]$. Each training point has a probability of $p=0.2$ to suffer from the noise.
> > > > > > >
> > > > > > > **Experimental results.** We first present the detailed results of the illustrative example that we pointed out in our above theoretical response and Appendix C. Then, to further verify the robustness of LTR losses, we vary the hyper-parameters of noise to create a heavier noise and examine the trained models.
> > > > > > >
> > > > > > > In Figure 3 in Appendix C (also available at https://anonymous.4open.science/r/Offline-RaM-7FB1/theory_simulation/Fig.pdf), we visualize the ground-truth function, training data, and the linear models trained with MSE and RankCosine. We can observe that the model learned from MSE exhibits a negative correlation, but the model learned from RankCosine can demonstrate a positive correlation.
> > > > > > >
> > > > > > > To further verify the robustness of ranking losses, we increase the dataset size to 100 and vary the scale of noise $\alpha\in$ { $10, 15, 20, 50, 100$ } while the probability of adding noise is fixed at $p=0.2$. We report the calculated values of $w$ learned with MSE (denoted as $w _ {MSE}$) and that learned with RankCosine (denoted as $w _ {RankCosine}$) according to different $\alpha$s in the following table.
> > > > > > >
> > > > > > > | Noise scale  $\alpha$ | $w _ {MSE}$ | $w _ {RankCosine}$ |
> > > > > > > |------------------------|-------------|--------------------|
> > > > > > > | 10                     | -1.68       | 0.88               |
> > > > > > > | 15                     | -0.75       | 0.90               |
> > > > > > > | 20                     | -2.98       | 0.74               |
> > > > > > > | 50                     | -14.99      | 0.94               |
> > > > > > > | 100                    | -10.05      | 0.98               |
> > > > > > >
> > > > > > > From the above table, all values of $w _ {RankCosine}$ are positive while those of $w _ {MSE}$ are all negative and become substantially worse when the scale of noise $\alpha$ goes larger, which demonstrates the stronger stability of the LTR loss against heavy-tailed noise with different strengths.
> > > > > > >
> > > > > > > We also vary the probability of adding noise $p\in$ { $0.1, 0.2, \cdots, 1.0$ } while the scale of noise is fixed at $\alpha=15$. The corresponding values of $w$ are listed as follows.
> > > > > > >
> > > > > > > | Noise  probability $p$ | $w _ {MSE}$ | $ w_ {RankCosine}$ |
> > > > > > > |-------------------------|-------------|--------------------|
> > > > > > > | 0.1                     | 1.61        | 0.86               |
> > > > > > > | 0.2                     | -0.75       | 0.90               |
> > > > > > > | 0.3                     | -4.53       | 1.01               |
> > > > > > > | 0.4                     | -8.46       | 0.88               |
> > > > > > > | 0.5                     | -7.76       | 1.02               |
> > > > > > > | 0.6                     | -10.73      | 0.95               |
> > > > > > > | 0.7                     | -12.87      | 0.84               |
> > > > > > > | 0.8                     | -17.28      | 0.98               |
> > > > > > > | 0.9                     | -20.21      | 0.98               |
> > > > > > > | 1.0                     | -22.66      | 0.95               |

---

> > > > > > > ### Author Response · Authors · 2024-11-23
> > > > > > > **Thanks and response (2/2)**
> > > > > > >
> > > > > > > From the results, only when the noise probability $p=0.1$, $w _ {MSE}$ is positive, while in other situations it is negative and it becomes quite bad as $p$ increases. In contrast, $w _ {RankCosine}$ remains a positive value near 1 as the noise probability $p$ increases from 0.1 to 1, showing impressive robustness against such heavy-tailed noise with wide coverage.
> > > > > > >
> > > > > > > The above results strongly demonstrate the robustness of pairwise ranking loss (i.e., RankCosine) over MSE on the ranking performance in a scenario where $y$ suffers from a heavy-tailed noise. Combining with the stated equivalence of an order-preserving surrogate model shown in Theorem 1, the ranking loss is suitable for offline MBO due to its more robust ranking performance.
> > > > > > >
> > > > > > > We have included these experimental results in Appendix C.2. With these additional results, we believe that the readers could have a better understanding of the advantage of LTR losses.
> > > > > > >
> > > > > > > **We hope that our response has addressed your concerns, but if we missed anything please let us know. Thank you again for your valuable suggestions and timely response.**

---

> > > > > > > > ### Comment · Reviewer_9jkq · 2024-11-23
> > > > > > > >
> > > > > > > > Thank you for providing the detailed computational experiment and explanation. The computational experiment demonstrates the advantages of using LTR losses. It highlights that training with MSE tends to place excessive weight on absolute values, making it prone to overfitting outliers. In contrast, LTR losses focus solely on the ordering among the values, reducing the influence of outliers and ensuring robustness.
> > > > > > > > Based on my understanding, this work appears to make a significant contribution to the relevant literature, and so I have increased my score.
> > > > > > > >
> > > > > > > > I do have one additional question: The use of LTR losses has also been proposed in a different field called 'predict-then-optimize' [1]. While the two topics may seem distant at first glance, they appear to share a similar motivation.
> > > > > > > >  [1] also aims to predict parameters for an optimization problem in a way that preserves the correct order of feasible solutions. Could you comment on whether there are parallels between your approach and the work [1]?
> > > > > > > >
> > > > > > > > 1: Mandi, J., Bucarey, V., Tchomba, M. M. K., & Guns, T. (2022, June). Decision-focused learning: Through the lens of learning to rank. In International conference on machine learning (pp. 14935-14947). PMLR.

---

> > > > > > > > > ### Author Response · Authors · 2024-11-25
> > > > > > > > > **Thanks for your appreciation!**
> > > > > > > > >
> > > > > > > > > Thanks for your appreciation! We are glad to hear that you recognize our contribution to the relevant field.
> > > > > > > > >
> > > > > > > > > As for the use of LTR losses in the field of “predict-then-optimize” [1] you mentioned, both our approach and [1] use LTR techniques to address domain-specific problems, but our work differs substantially in problem modeling, methodology, and theoretical underpinnings. For example, the major difference in modeling is the different interpretations of *items to be sorted,* which causes differences in loss calculation, training method, and model usage.
> > > > > > > > >
> > > > > > > > > In fact, as a fundamental technique of machine learning, LTR has been adopted to many fields to address the specific problems. For example,
> > > > > > > > >
> > > > > > > > > - In preference-based reinforcement learning, [2] defines a preference oracle to measure the total order and uses pairwise ranking loss to train a reward model for the sparse-reward environments.
> > > > > > > > > - In language model alignment, [3] and [4] efficiently deal with the human preference rankings from pairwise and listwise ranking perspectives, respectively.
> > > > > > > > >
> > > > > > > > > However, our work differs from these mentioned works in both motivation and methodology. We focus on offline MBO and investigate the root cause of the OOD issue, which is widely-studied in this field but still remains. We provide a systematic analysis of the OOD issue, propose the AUPCC metric for quantification, develop a ranking-based framework, and verify its effectiveness through theoretical analysis and comprehensive experiments. We believe that our work makes a significant contribution to the offline MBO field.
> > > > > > > > >
> > > > > > > > > **We have included a discussion of works that also leverage LTR techniques to advance their respective fields in Appendix A.2. Thank you again for your appreciation and constructive suggestions!**
> > > > > > > > >
> > > > > > > > > ### References
> > > > > > > > >
> > > > > > > > > [1] Jayanta Mandi, Vı́ctor Bucarey, Maxime Mulamba Ke Tchomba, Tias Guns. Decision-focused learning: Through the lens of learning to rank. In Proceedings of the 39th International Conference on Machine Learning (ICML’22), pp. 14935–14947, Baltimore, MD, 2022.
> > > > > > > > >
> > > > > > > > > [2] Farzan Memarian, Wonjoon Goo, Rudolf Lioutikov, Scott Niekum, and Ufuk Topcu. Self-supervised online reward shaping in sparse-reward environments. In 2021 IEEE/RSJ International Conference on Intelligent Robots and Systems (IROS‘21), pp. 2369–2375, Prague, Czech Republic, 2021.
> > > > > > > > >
> > > > > > > > > [3] Feifan Song, Bowen Yu, Minghao Li, Haiyang Yu, Fei Huang, Yongbin Li, and Houfeng Wang. Preference ranking optimization for human alignment. In Proceedings of the 38th AAAI Conference on Artificial Intelligence (AAAI’24), pp. 18990–18998, Vancouver, Canada, 2024.
> > > > > > > > >
> > > > > > > > > [4] Tianqi Liu, Zhen Qin, Junru Wu, Jiaming Shen, Misha Khalman, Rishabh Joshi, Yao Zhao, Mo-hammad Saleh, Simon Baumgartner, Jialu Liu, Peter J. Liu, and Xuanhui Wang. LiPO: Listwise preference optimization through learning-to-rank. arXiv:2402.01878, 2024.

---

> ### Author Response · Authors · 2024-11-18
> **Response by the authors (2/3)**
>
> ### Q3 Does not the proposed approach require tricks like adversarial weight perturbation?
>
> Thanks for your comment. We want to make some clarifications. We do not claim that the proposed approach does not require such tricks like perturbation. We just claim that learning with ranking aligns better with offline MBO than learning with MSE. Generalization tricks in prior approaches can also be applied to our method.
>
> As discussed in Section 3.1 and 3.2, we show that it is enough for offline MBO by solely focusing on model's ability to preserve order. Thus, we propose to learn the ranking directly. Note that ranking is a weaker condition since it only commands for correct orders, while minimizing MSE argues for both order preserving and value approximating. The results in Appendix F.5 and Table 11 show that our method can improve ranking performance even in OOD regions.
>
> However, our proposed ranking-based framework also suffers from OOD issue, where the surrogate model might erroneously predicts the order relationship in OOD regions. Note that there have been various efforts to mitigate the OOD issue in offline MBO (e.g., adding sensitivity regularization terms [1]), but such tricks are done on the regression-based framework. Although our proposed framework is based on ranking, some of those tricks can also be applied. In Table 3, we examined this by simply replacing the MSE loss of prior methods with our best-performing ranking loss, ListNet, and incorporating output adaptation, which lead to better performance. This demonstrates the versatility of ranking loss and the possibility to apply prior methods in offline MBO to our proposed ranking framework.
>
> COMBO [2], i.e. adversarial weight perturbation as you mentioned, is an approach for offline RL, which is different from offline MBO. Specifically, offline MBO can be viewed as a one-step decision-making procedure, where we propose only one batch of design $\mathbf{x}$ for evaluation, while offline RL is a multi-step decision-making procedure to maximize a long-term discounted reward of a policy $\pi$. This causes differences between offline RL and offline MBO from many perspectives, as discussed in Table 1 of [3]. Thus, COMBO cannot be directly applied to our framework. However, we have combined our framework with other applicable techniques in offline MBO in Table 3 of our paper, like ensembling weak learners (i.e., Tri-Mentoring [4]), learning from policies (i.e., PGS [5]), and matching the gradient (i.e., MATCH-OPT [6]), by simply replacing the core MSE term in their loss function by ranking loss and incorporating output adaptation, which demonstrates both the versatility of the ranking loss and the compatibility of these techniques with our ranking-based framework.
>
>
> ### Q4 Unclear how using an LTR loss alone could eliminate the need for approaches
>
> Thank you for your comments. As our reply to Q3, we do not claim that using an LTR loss alone or using the ranking framework could eliminate the need for such approaches. As mentioned above, we combined our ranking framework with some recent approaches in offline MBO (e.g., Tri-Mentoring [4], PGS [5], and MATCH-OPT [6]) shown in Table 3, and we also explained why not combining it with other works in Appendix E.5. The superior performance led by these straightforward combination has demonstrated the possibility to apply prior methods in offline MBO to our proposed ranking framework, and how to combine them more effectively is an interesting future work.
>
>
> ### Q5 How to guarantee that the ranking will generalize to out-of-distribution data?
>
> Thank you for your insightful question. In this work, we point out that the model's ability to preserve order in OOD regions is crucial for offline MBO, instead of focusing only on OOD-MSE as in previous works. The OOD-AUPCC results in Table 11 show that using LTR techniques is more effective to generalize well on OOD data than regression-based methods. However, due to the inherent nature of the limited coverage of the offline dataset (which usually covers a narrow manifold [7]), the OOD issue is inevitable, and also exists for the ranking-based framework. In the future, it is expected to improve the model's generalization ability to properly rank the samples in OOD regions, e.g., by utilizing previous techniques in regression-based offline MBO as mentioned before.
>
>
> **We hope that our response has addressed your concerns, but if we missed anything please let us know.**

---

> ### Author Response · Authors · 2024-11-18
> **Response by the authors (3/3)**
>
> ### References
>
> [1] Manh Cuong Dao, Phi Le Nguyen, Thao Nguyen Truong, Trong Nghia Hoang. Boosting offline optimizers with surrogate sensitivity. In Proceedings of the 41st International Conference on Machine Learning (ICML'24), pp. 10072-10090, Vienna, Austria, 2024.
>
> [2] Tianhe Yu, Aviral Kumar, Rafael Rafailov, Aravind Rajeswaran, Sergey Levine, Chelsea Finn. COMBO: Conservative offline model-based policy optimization. In Advances in neural information processing systems 34 (NeurIPS'21), pp. 28954-28967, Virtual, 2021.
>
> [3] Han Qi, Yi Su, Aviral Kumar, Sergey Levine. Data-driven offline decision-making via invariant representation learning. In Advances in neural information processing systems 35 (NeurIPS'22), pp. 13226-13237, New Orleans, LA, 2022.
>
> [4] Can (Sam) Chen, Christopher Beckham, Zixuan Liu, Xue (Steve) Liu, Chris Pal. Parallel-mentoring for offline model-based optimization. In Advances in Neural Information Processing Systems 36 (NeurIPS'23), pp. 76619-76636, New Orleans, LA, 2023.
>
> [5] Yassine Chemingui, Aryan Deshwal, Trong Nghia Hoang, Janardhan Rao Doppa. Offline model-based optimization via policy-guided gradient search. In Proceedings of the 38th AAAI Conference on Artificial Intelligence (AAAI'24), pp. 11230–11239, Vancouver, Canada, 2024.
>
> [6] Minh Hoang, Azza Fadhel, Aryan Deshwal, Jana Doppa, Trong Nghia Hoang. Learning surrogates for offline black-box optimization via gradient matching. In Proceedings of the 41st International Conference on Machine Learning (ICML'24), pp. 18374-18393, Vienna, Austria, 2024.
>
> [7] Aviral Kumar, Sergey Levine. Model inversion networks for model-based optimization. In Advances in Neural Information Processing Systems 33 (NeurIPS'20), pp. 5126–5137, Virtual, 2020.

---

### Meta-Review · Area_Chair_nmcd · 2024-12-19

**Metareview:**

The paper considers the problem of offline model based optimization. The key idea is to use learning to rank loss instead of mean squared error loss for training surrogate models. Although this point has already been discussed earlier in MatchOpt, the paper does more thorough and comprehensive analysis. I think the community will benefit from this paper. Therefore, I recommend acceptance.

**Additional Comments On Reviewer Discussion:**

Overall, all reviewers liked the key idea of learning to rank for this problem setting. Reviewers 9jkq and ChpN had concerns about metrics and evaluation procedure which were addressed in the author rebuttal. Some reviewers also pointed out the lack of novelty in the theory of the paper. I request the authors to consider incorporating changes in the paper based on these comments.

---

### Decision · Program_Chairs · 2025-01-22

Accept (Poster)